# L-Drive: Beyond a Single Mapping—Latent Context Drives Time Series Forecasting

**Fan Zhang** [1]   **Shijun Chen** [1]   **Hua Wang** [2]

## Abstract

Mainstream methods for multivariate time-series forecasting largely follow the Direct-Mapping paradigm. They learn a unified mapping from history to the future in the observation space to fit value-level dependencies. However, real-world systems often undergo distribution shifts and regime changes. In such cases, a unified mapping can exhibit response lag around turning points, causing error accumulation within the switching window and reducing forecasting reliability. To address this issue, we propose L-Drive, a change-aware forecasting framework. L-Drive introduces a Latent-Context, to explicitly characterize high-level dynamics evolving over time, and uses gating to modulate increment representations. This provides more timely change cues and improves adaptation to changing segments. In addition, it incorporates patch-shared relative positional basis functions to strengthen intra-segment structural modeling and reduce overfitting caused by absolute-position memorization. Extensive experiments validate the effectiveness of L-Drive and show a better overall trade-off between forecasting accuracy and computational efficiency.

## 1. Introduction

Time series forecasting is one of the core tools for understanding and driving real-world systems (Wu et al., 2025; Li et al., 2026; Liu et al., 2026a). It is widely used in power load (Mroueh et al., 2025; Jain & Gupta, 2024) and energy scheduling (Dong et al., 2023), transportation and logistics, finance and risk management (Chen et al., 2024b), industrial equipment monitoring (Pinciroli Vago et al., 2024), meteorology and environment (Waqas et al., 2024), and public

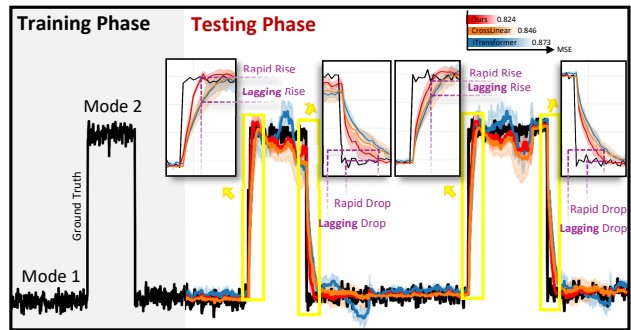

*Figure 1.* Comparison on synthetic data (original settings for each baseline). Our model adapts faster, reducing lag around switches.

health (Chhabra et al., 2024). In these scenarios, accurate multi-step forecasting not only directly affects resource allocation and decision quality, but also determines system safety and robustness under uncertainty.

At present, many mainstream approaches for multivariate time series forecasting can be abstracted as learning a unified value-level mapping from historical observations to the future horizon in the observation space, aiming to capture cross-variable correlations and temporal dependencies at the level of observed values (Chen et al., 2023). We refer to this input-output formulation as Direct-Mapping. From a structural perspective, they can be roughly summarized along two design dimensions. The first dimension is cross-variable interaction (Qiu et al., 2025a; Kim et al., 2025; Zhao & Shen, 2024; Chen et al., 2024a). One class of methods adopts channel-mixing strategies to jointly model variable–time relationships, while another class prefers channel independence and encodes temporal dependencies for each variable separately. The second dimension is variable conditioning (Lim & Zohren, 2021). By partitioning variables into endogenous and exogenous variables, these methods achieve conditional modeling that integrates both types of variables. In addition, from a representation perspective, some studies apply domain transformations to the sequence and introduce time–frequency representations to explicitly characterize multi-scale features (Zhang et al., 2025; Yi et al., 2025; Yang et al., 2024; Lange et al., 2021). It should be emphasized that, regardless of how variables interact structurally

[1] Shandong Technology and Business University, Yantai, Shandong, China [2] Ludong University, Yantai, Shandong, China. Correspondence to: Hua Wang <hua.wang@ldu.edu.cn>.

*Proceedings of the 43rd International Conference on Machine Learning*, Seoul, South Korea. PMLR 306, 2026. Copyright 2026 by the author(s).

or whether domain transformations are introduced, these methods still mainly focus on correlations and dependency structures of observed values. They often perform well when the dynamic patterns of the sequence remain relatively stable. However, real-world systems rarely operate under a single regime for a long time. Real-world systems are often affected by policy adjustments, environmental changes, and unexpected events. When such events occur, the fluctuation patterns, correlation structures, and even the rate of change of the sequence may shift (Ryan et al., 2025; Dixit & Jain, 2023). Under this Direct-Mapping formulation, when dynamic regimes switch, predictions can exhibit a form of lagged extrapolation. That is, during several time steps after the switch, the model still tends to extrapolate based on the local trend and level learned before the switch. This leads to insufficient responses to sudden level shifts or slope changes. As a result, the prediction curve underestimates the magnitude around turning points and becomes temporally misaligned with the true sequence.

To more clearly characterize this lag phenomenon, we design a synthetic dataset with extreme dynamic switching. Under the base pattern, Mode 1, the sequence evolves with a linear upward trend of small slope and added random noise. At irregular time points, the sequence instantaneously switches to Mode 2. In Mode 2, the local slope is similar to that of Mode 1, but the intercept is shifted upward by a fixed amount relative to Mode 1. After a certain duration, the sequence instantaneously switches back to Mode 1. Such jumps appear randomly in both training and testing. The experimental results shown in Figure 1 demonstrate that two mainstream baseline models, the MLP-based CrossLinear and the Transformer-based iTransformer, both exhibit systematic level biases near the switching points between Mode 1 and Mode 2. The errors show a slowly decaying tail, which reflects a typical prediction lag.

The root cause lies in the fact that, before the change occurs, these models cannot explicitly distinguish at the prediction time whether the system is still following the current regime or is about to switch to a new one. Under a training paradigm that minimizes global average error, the models reinforce their reliance on the current local pattern using a large number of recent stable samples. They thus tend to learn a conservative extrapolation inertia, where a unified value-level mapping is used to fit observations across different historical phases. As a result, when a switch occurs, predictions are more likely to continue the pre-switch level and trend. Adjustments usually become evident only after information from the new regime is sufficiently reflected in the input window. Consequently, errors around the switching point persist for several steps and gradually decay, manifesting as lag and error accumulation. From another perspective, this lag can be viewed as an inevitable side effect of a recent-is-more-important weighted update

mechanism under dynamic switching. Adaptive expectations in economics (Chow et al., 2011) provide an intuitive statistical analogy. Expectations can be seen as a geometrically decaying weighted average of past observations, where recent information receives higher weight. This strategy is usually effective in stable environments, but under dynamic switching it naturally introduces inertia due to continued reliance on past history. Therefore, relying solely on a unified mapping in the observation space makes it difficult to update forecasting bases in a timely manner. Introducing an intermediate signal that can explicitly represent changes and drive rapid alignment to new regimes becomes key to alleviating this issue.

Based on this observation, we introduce Latent-Context (L-Context) to characterize dynamic patterns that evolve over time, and use it to modulate incremental representations, thereby improving the model's sensitivity and adaptation speed to regime switching. Specifically, we extract cues from historical sequences that reflect how the system changes, and selectively emphasize them through a gating mechanism. This allows the model to learn a time-evolving L-Context that represents the current change pattern. The core idea is to shift forecasting from using a unified mapping to cover all phases to scenario-aware modeling conditioned on L-Context. When system dynamics switch, the model can leverage this context to promptly update the basis for future changes, reducing lag and error accumulation within the switching window.

To more intuitively position L-Context, we compare it with two common perspectives in time series modeling. One focuses on explicitly characterizing repeatable temporal structures, such as trends and seasonality. The other models the intrinsic dynamics of a system over time through latent states. Specifically, L-Context does not learn a deterministic decomposed component. Instead, it represents a latent dynamic variation that is adaptively inferred from historical observations. It characterizes which type of change pattern the system is currently following and is used to modulate the forecasting process. This representation can capture periodic behaviors, as well as non-periodic regime switches and structural shifts. When a switch occurs, its main value lies in reducing lag and accelerating adaptation. Meanwhile, unlike the classical generative latent states in state space models, L-Context is not designed to generate observations and does not rely on explicit state space equations. Instead, it serves as a conditioning variable that drives forecasting, helping the predictor quickly align across different phases.

In terms of overall implementation, we combine L-Context with a patch based design and further introduce relative positional basis functions within each patch. These functions use learnable positional preference templates to characterize relative index structures inside a patch, thereby enhancing

the representation of local dynamic patterns. Overall, our method is built on a lightweight MLP framework and balances modeling capacity with computational efficiency. As shown in Figure 1, on the synthetic dataset our model can complete pattern alignment more quickly after a switch occurs. It significantly reduces the number of steps required for correction, thereby mitigating the accumulation of lag errors and improving adaptation to changes, which leads to a clear reduction in prediction bias. Furthermore, experiments on a wide range of real-world datasets show that the proposed model achieves stable and competitive forecasting performance with good generalization ability.

In summary, our contributions are as follows:

- We show that mainstream Direct-Mapping models tend to exhibit delayed responses at change points. Errors accumulate within the switching window and decay slowly, which increases overall forecasting error.

- We propose to extract latent context representations from historical observations to characterize the current change pattern and conditionally modulate the forecasting process.

- We introduce relative positional basis functions within each patch. By using learnable positional templates to explicitly model relative index structures, this design enhances local dynamic modeling ability without increasing modeling complexity.

## 2. Related Work

Time series modeling is one of the important research directions in the field of deep learning (Yang et al., 2026; Fu et al., 2025; Chen et al., 2026; 2025; Fu et al., 2026). Many excellent works have advanced time-series forecasting from diverse perspectives (Liu & Wang, 2024; Casolaro et al., 2023; Kim et al., 2025; Liu et al., 2026b). Some methods follow the Direct-Mapping paradigm. They learn a unified mapping from history to future in the observation space. The goal is to model time dependencies and cross-variable correlations at the value level.

Structural studies mainly focus on two dimensions. The first is cross-variable interaction: some methods adopt channel mixing (Tang & Zhang, 2025) to project multiple variables into a shared representation space and explicitly model inter-variable relations, while others prefer channel independence (Nie, 2022; Liu et al., 2023a), treating each variable as an individual series with shared parameters to mitigate overfitting from excessive coupling. The second dimension is variable conditioning (Wang et al., 2024; Zhou et al., 2025a), which distinguishes endogenous and exogenous variables to organize information flow and enhance conditional modeling.

At the representation level, several approaches introduce time–frequency signals via domain transformations to capture multi-scale patterns, such as leveraging frequency analysis and frequency-domain filtering for efficient modeling (Wang et al., 2025; Wu et al., 2022). Overall, regardless of architectural interaction choices or time–frequency transformations, these methods largely follow Direct-Mapping by learning a unified mapping centered on value-level dependencies.

Recent studies have increasingly focused on the generalization challenges caused by shift, forming a common line of work that starts from distribution changes at the statistical level (Liu et al., 2023c). These methods mitigate scale and bias differences across time segments through normalization, rescaling, or robustness mechanisms. A representative approach, RevIN (Kim et al., 2021), uses reversible normalization and denormalization to reduce performance degradation under distribution drift. TimeBridge (Liu et al., 2024) further discusses the trade-off between removing and preserving non-stationarity, arguing that short-term forecasting should suppress non-stationary fluctuations to avoid spurious correlations, whereas long-term forecasting needs to retain non-stationary components to capture long-term cointegration across variables, and proposes a divide-and-conquer framework accordingly. Koopa (Liu et al., 2023b) leverages Koopman theory to represent and propagate time-varying dynamics, enhancing the modeling of non-stationary evolution. In addition, for multimodal alignment, BALM-TSF (Zhou et al., 2025b) addresses the imbalance between textual semantics and time-series representations in LLM-based TSF and improves forecasting via more balanced cross-modal alignment. For generative forecasting, mr-Diff (Shen et al., 2024) combines multi-scale decomposition with diffusion denoising to model predictive distributions and capture multi-scale patterns.

It is important to note that, unlike the above non-stationarity treatments, L-Drive focuses on regime switches and change patterns in the underlying dynamics. It explicitly constructs time-updated change cues to reduce response lag around switching points. Response lag is related to non-stationarity and distribution shift, but it refers to a more specific temporal phenomenon: the delayed adjustment of predictions after local dynamic changes, such as regime switches, level shifts, and trend turning points. Existing non-stationarity treatments mainly address statistical mismatch, normalization bias, or long-term distributional variation, whereas L-Drive focuses on reducing the temporal delay of prediction responses around switching regions. In this sense, our work complements these approaches by introducing variation-related context to modulate forecasting updates under local dynamic changes.

## 3. Motivation

To understand the response lag that is commonly observed in Direct-Mapping models at distribution shifts and trend turning points, we analyze the local behavior of their prediction trajectories. Consider a class of predictors that can be abstracted into the Direct-Mapping form, whose input-output relationship can be uniformly expressed as:

$$\hat{y}_t = f_\theta(x_{1:t}), \tag{1}$$

where $x_{1:t}$ denotes the historical observations up to time $t$, and $f_\theta(\cdot)$ is a parameterized forecasting mapping. For simplicity, we denote the prediction output as $\hat{y}_t$.

During actual forecasting, the model typically takes a fixed historical length as input and outputs a prediction interval through a single forward propagation. Near the regime switch, pre-switch information gradually exits as the window slides, while post-switch information gradually enters, causing neighboring predictions to exhibit local inertia. Based on this phenomenon, we use the following reduced-form relation to summarize the inertia term in neighboring predictions and the update term induced by the current input and the detailed proof is provided in Appendix A:

$$\hat{y}_t \approx \rho\hat{y}_{t-1} + g_\theta(x_t), |\rho| < 1. \tag{2}$$

More generally, Appendix A yields a time-varying local coefficient $\rho_t$; in a small operating region we assume $|\rho_t| \le L < 1$ and use a representative constant $\rho$ in Equation (2) for notational simplicity. The term $g_\theta(x_t)$ summarizes the update driven by the current observation. Appendix A quantitatively evaluates how well Equation (2) approximates the local prediction behavior of single-forward-pass baselines around switching events.

The above expression reveals a key fact: even if the model does not explicitly adopt an autoregressive or distributed lag structure, its prediction trajectory near the switching point may still reflect the joint effect of partial retention of the previous prediction level and updates induced by current input information. As a result, its deployed recursive dynamics typically exhibits continuing extrapolation behavior in locally contractive regimes. Historical predictions remain in the system through the coefficient $\rho$, and they pull the prediction trajectory when a structural change occurs. This structure is mathematically consistent with the classical adaptive expectation model (Chow et al., 2011), where the current prediction is composed of a retention term from the historical prediction and an update term driven by new observations:

$$P_t^e = \rho P_{t-1}^e + (1 - \rho)P_{t-1}, \rho \in (0, 1), \tag{3}$$

here the adaptive-expectations form corresponds to the common monotone case, thus $\rho \in (0, 1)$.

This indicates that prediction inertia is a generic byproduct of recursive deployment in locally stable (contractive) regimes. Therefore, relying solely on a unified mapping in the observation space makes it difficult to quickly break free from the constraints of historical states. As a result, the model produces systematic response lag at distribution shifts or dynamic turning points. This limitation forms the core motivation for the method proposed in this paper.

## 4. Methodology

As shown in the framework in Figure 2, L-Drive is composed of two modules, the L-Context Generator and the Struct-Aided Predictor. The following sections provide detailed descriptions of the implementation of each module and the code is avaliable at https://github.com/ShijunChen01/L-Drive.

### 4.1. L-Context Generator

To improve the response speed of the model to distribution shifts and dynamic turning points, we no longer rely solely on a unified mapping in the observation space. Instead, we introduce explicit modeling of changes. Specifically, we reduce changes to their most basic components, namely the increments between adjacent time points. Therefore, first-order differencing is the most direct way to model change. However, the differencing operation tends to amplify high-frequency disturbances and introduce noise. To make the resulting change signal more stable and controllable, we first preprocess the original sequence. Given the input sequence $x = \{x_t\}_{t=1}^T, x_t \in R^V$ with length $T$ and $V$ feature channels, we perform standard normalization:

$$x'_t = \frac{x_t - \mathbb{E}[x]}{\sqrt{\mathbb{E}[(x - \mathbb{E}[x])^2] + \varepsilon}}, \tag{4}$$

here, $\varepsilon$ is a numerical stability term used to avoid potential numerical instability.

Then, considering that the same latent change may appear with different delays across different channels, meaning that the same event can manifest as responses over different time intervals in each variable, we apply a one-dimensional convolution along the time dimension to align the changes. The convolution output is not used to replace the original sequence. Instead, it is treated as the aligned multi-channel sequence data and added to the original data branch in a residual manner. For the preprocessed sequence, we further compute the first-order difference between adjacent time steps. At the same time, we set $(\mathcal{D}x')_1$ to zero as the natural boundary condition for the differencing process. This operation maintains sequence length alignment while avoiding imposing redundant prior constraints on the magnitude or

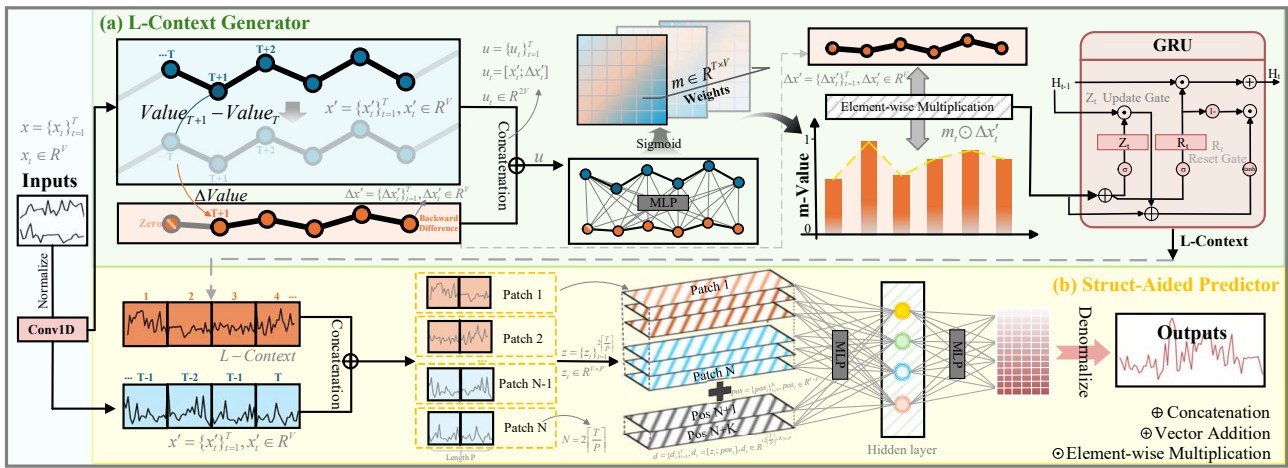

*Figure 2.* Overview of L-Drive. It consists of two key components: (a) L-Context Generator and (b) Struct-Aided Predictor.

direction of change at the initial time step:

$$\Delta x' = \mathcal{D}(x'),$$

$$(\mathcal{D}x')_t = \begin{cases} 0, & t = 1, \\ x'_t - x'_{t-1}, & t = 2, \dots, T. \end{cases} \quad (5)$$

It should be noted that normalization mainly provides global-scale stabilization, and it cannot eliminate local spikes or instantaneous high-frequency disturbances that appear after differencing. Therefore, we further introduce a gating mechanism in the differenced domain to adaptively filter the incremental change at each step.

$$u = \{u_t\}_{t=1}^T, u_t \in R^{2V},$$
$$u_t = [x'_t; \Delta x'_t],$$
$$= [x'_{t,1} \dots x'_{t,V}, \Delta x'_{t,1} \dots \Delta x'_{t,V}]^T, \quad (6)$$
$$m_t = g(u_t),$$
$$h_t = m_t \odot \Delta x'_t,$$

here, $g(\cdot)$ is an MLP-based gating network implemented as a single-layer network with a sigmoid activation. It projects the input from $2V$ to $V$, ensuring dimensional alignment for subsequent element-wise multiplication.

The gating mechanism is driven by both the original observations and the change signals. This allows the model to strengthen truly important changes in the context of the current state, while suppressing inconsistent or weakly related incremental components, thereby reducing the high-frequency noise that differencing may introduce. The denoised change sequence then needs to be integrated into a stable state. A sequence is essentially a dynamic quantity that evolves over time, and recursive integration can preserve temporal relationships completely. Therefore, we introduce the GRU, utilizing its simple gated recursive units along with adaptive control for memory and forgetting, ultimately obtaining the hidden state sequence L-Context at

each time step of the GRU, which serves as a basis for fast updates in subsequent predictions.

$$L - Context = GRU(h_t). \quad (7)$$

### 4.2. Struct-Aided Predictor

After concatenating L-Context with the preprocessed original sequence along an additional fusion dimension, we obtain $l \in R^{V \times 2 \times T}$ as the final input for forecasting. To improve generalization, we perform non-overlapping patch partitioning of the fused sequence along the time dimension. The patch size is $P$, the number of patches for each of the two fused components is $\lceil \frac{T}{P} \rceil$, and the patch sequence is defined as $z = \{z_t\}_{t=1}^{2\lceil \frac{T}{P} \rceil}, z_t \in R^{V \times P}$. Importantly, after patching, we introduce relative positional basis functions within each patch. Specifically, for each variable, we learn $K$ relative positional bases of length $P$, denoted as $pos = \{pos_t\}_{t=1}^K, pos_t \in R^{V \times P}$. This allows the model to obtain learnable reference mappings of positional preference within each patch, thereby strengthening the expression of internal structure within the patch.

In change-aware forecasting, regime switches, level shifts, and slope changes may occur at arbitrary positions within the input window. Standard absolute positional encodings may encourage the predictor to associate change patterns with specific locations observed during training, rather than learning the local evolution of the changes themselves. To mitigate this issue, the Struct-Aided Predictor adopts patch-wise relative positional bases. Compared with absolute positional encodings, this design is independent of global time indices and shared across patches, allowing the model to learn local structural preferences within each patch instead of memorizing specific absolute time steps. This reduces position-specific overfitting and helps the variation-related cues encoded by L-Context be translated more stably into

responses to local dynamics. Then, we apply an MLP-based mapping to the combined patches of data and positional bases to obtain patch-level representations:

$$
\begin{aligned}
&d = \{d_t\}_{t=1}^V, d_t \in R^{(2\lceil \frac{T}{P} \rceil + K) \times P}, \\
&d_t = [z_t; pos_t] = [z_{t,1} \ldots z_{t,2\lceil \frac{T}{P} \rceil}, pos_{t,1} \ldots pos_{t,K}]^T, \\
&y' = \{y'_t\}_{t=1}^V, y'_t = g_1(d_t), y'_t \in R^{(2\lceil \frac{T}{P} \rceil + K) \times Q},
\end{aligned}
\tag{8}
$$

here, $g_1(\cdot)$ is an MLP-based gating network and $Q$ denotes the size of the hidden layer.

Compared with a global mapping, this hierarchical structure of local mappings followed by recombination is equivalent to constructing a composite mapping composed of multiple segmental subfunctions. It can capture heterogeneous patterns and local shape differences in different time intervals at a finer granularity, thereby significantly improving expressive power. Then, the predictor merges all patches and maps them to the final prediction. Finally, the result is transformed back to the original scale through inverse normalization, completing the output forecast of length $H$:

$$
\begin{aligned}
&\hat{y} = \{\hat{y}_t\}_{t=1}^H, \hat{y}_t \in R^V, \\
&\hat{y}_t = y'_t(\sqrt{E[(x - \mathbb{E}[x])^2]} + \varepsilon) + \mathbb{E}[x].
\end{aligned}
\tag{9}
$$

### 4.3. Theoretical Implication of L-Context

We have obtained the inertial form as shown in Equation (2). It can be rewritten in the standard EWMA form:

$$
\begin{aligned}
&g_t \triangleq \frac{g_\theta(x_t)}{1 - \rho}, \\
&\hat{y}_t \approx \rho \hat{y}_{t-1} + (1 - \rho)g_t.
\end{aligned}
\tag{10}
$$

By iteratively expanding Equation (10):

$$
\hat{y}_t \approx \rho^K \hat{y}_{t-K} + (1 - \rho) \sum_{k=0}^{K-1} \rho^k g_{t-k}.
\tag{11}
$$

We introduce L-Context and use it to provide an increment-related signal. We write it as

$$
\Delta \hat{g}_t = \varphi_\theta(L - Context),
\tag{12}
$$

this increment-related interpretation of L-Context is further supported in Appendix C, where we provide both a theoretical bounded-error derivation and an empirical distance-correlation analysis.

Based on Equation (10), we view L-Context as primarily contributing an increment-like correction term:

$$
\hat{y}_t^L = \rho \hat{y}_{t-1}^L + (1 - \rho)g_t + \rho \Delta \hat{g}_t.
\tag{13}
$$

In practice,
$$
\Delta \hat{g}_t = \Delta g_t + \varepsilon_t,
\tag{14}
$$

where $\varepsilon_t$ denotes the increment estimation error.

We define the tracking error as

$$
e_t \triangleq \hat{y}_t^L - g_t.
\tag{15}
$$

It can be shown, as detailed in Appendix D, that

$$
\begin{aligned}
&e_t = \rho e_{t-1} + \rho \varepsilon_t, \\
&|e_t| \leqslant \rho^K |e_{t-K}| + \rho \sum_{k=0}^{K-1} \rho^k |\varepsilon_{t-k}|.
\end{aligned}
\tag{16}
$$

If the increment estimation error is bounded as $|\varepsilon_t| \leqslant \bar{\varepsilon}$, then the long-term residual lag is upper bounded by:

$$
\limsup_{t \to \infty} |e_t| \leq \frac{\rho}{1 - \rho} \bar{\varepsilon}.
\tag{17}
$$

L-Context makes the lag no longer mainly determined by the trailing effect of the model's inertial structure. Instead, it is primarily governed by the increment estimation error $\varepsilon_t$ of L-Context. The smaller $\bar{\varepsilon}$ becomes, the smaller the residual lag will be in theory. Moreover, the residual lag admits an explicit upper bound given by $\frac{\rho}{1-\rho}\bar{\varepsilon}$.

## 5. Experiments

This section describes the experimental setup and metrics, analyzes the results, and further validates each module's contribution through targeted component experiments.

### 5.1. Experimental Setting

For long-term forecasting, we use seven classic datasets in the time series field, namely ETT (ETTm1, ETTm2, ETTh1, ETTh2), Electricity (ECL), Weather, and Solar_Energy (Solar) (Wu et al., 2021; Liu et al., 2023a). We use a look-back window of 96 and prediction lengths of 96, 192, 336, 720. For short-term forecasting, we use the EPF dataset (Lago et al., 2021) with a lookback window of 168 and a prediction length of 24. We compare both settings with nine widely used baselines: TimeBridge (Liu et al., 2024), CrossLinear (Zhou et al., 2025a), FilterTS (Wang et al., 2025), PatchMLP (Tang & Zhang, 2025), iTransformer (Liu et al., 2023a), TimeXer (Wang et al., 2024), PatchTST (Nie, 2022), TimesNet (Wu et al., 2022), and DLinear (Zeng et al., 2023). Specifically, CrossLinear, PatchMLP, and DLinear are MLP-based. TimeBridge, TimeXer, iTransformer and PatchTST are Transformer-based. TimesNet is CNN-based, and FilterTS is time–frequency-based. We report Mean Squared Error (MSE) and Mean Absolute Error (MAE) as evaluation metrics (Zhou et al., 2021).

### 5.2. Results Analysis

As shown in Table 1, our method delivers consistently strong performance across all eight datasets. Among the 16 evaluated cases, it achieves the best results in 12 cases and the

*Table 1.* Averaged results on the forecasting task. **Bold** indicates the best results, and underlining indicates the second-best results.

| Datasets | L-Drive (Ours) | | TimeBridge (2025) | | CrossLinear (2025) | | FilterTS (2025) | | PatchMLP (2025) | | iTransformer (2024) | | TimeXer (2024) | | PatchTST (2023) | | TimesNet (2023) | | DLinear (2023) | |
|---|---|---|---|---|---|---|---|---|---|---|---|---|---|---|---|---|---|---|---|---|
| | MSE | MAE | MSE | MAE | MSE | MAE | MSE | MAE | MSE | MAE | MSE | MAE | MSE | MAE | MSE | MAE | MSE | MAE | MSE | MAE |
| ETTh1 | **0.424** | **0.430** | 0.438 | 0.434 | 0.431 | 0.432 | 0.433 | **0.430** | 0.463 | 0.447 | 0.463 | 0.454 | 0.437 | 0.437 | 0.469 | 0.455 | 0.458 | 0.450 | 0.456 | 0.452 |
| ETTh2 | **0.366** | **0.396** | 0.375 | 0.397 | 0.368 | **0.396** | 0.372 | **0.396** | 0.395 | 0.412 | 0.383 | 0.407 | 0.368 | **0.396** | 0.387 | 0.407 | 0.414 | 0.427 | 0.559 | 0.515 |
| ETTm1 | **0.371** | 0.393 | 0.383 | **0.391** | **0.371** | 0.393 | 0.385 | 0.396 | 0.392 | 0.400 | 0.407 | 0.411 | 0.382 | 0.397 | 0.387 | 0.400 | 0.400 | 0.406 | 0.403 | 0.407 |
| ETTm2 | 0.273 | **0.320** | 0.284 | 0.322 | **0.272** | 0.321 | 0.276 | 0.337 | 0.287 | 0.331 | 0.290 | 0.334 | 0.274 | 0.322 | 0.281 | 0.326 | 0.291 | 0.333 | 0.350 | 0.401 |
| ECL | **0.169** | **0.266** | 0.175 | 0.267 | 0.173 | 0.270 | 0.180 | 0.272 | 0.196 | 0.288 | 0.178 | 0.270 | 0.171 | 0.270 | 0.205 | 0.290 | 0.193 | 0.295 | 0.212 | 0.300 |
| Weather | **0.235** | **0.267** | 0.254 | 0.271 | 0.238 | 0.269 | 0.245 | 0.274 | 0.248 | 0.275 | 0.258 | 0.278 | 0.241 | 0.271 | 0.259 | 0.281 | 0.259 | 0.287 | 0.265 | 0.317 |
| Solar | 0.230 | 0.262 | **0.227** | **0.236** | 0.228 | 0.260 | 0.248 | 0.283 | 0.277 | 0.293 | 0.233 | 0.262 | 0.236 | 0.275 | 0.270 | 0.307 | 0.301 | 0.319 | 0.330 | 0.401 |
| EPF | **0.168** | **0.207** | 0.209 | 0.245 | 0.172 | 0.210 | 0.190 | 0.233 | 0.191 | 0.232 | 0.189 | 0.227 | 0.175 | 0.216 | 0.183 | 0.229 | 0.184 | 0.225 | 0.218 | 0.275 |

second-best results in the remaining 2 cases. Compared with MLP-based baselines, our method reduces MSE and MAE by up to 6.9% and 5.3%, respectively, over CrossLinear on Electricity, and achieves up to 49.3% lower MSE and 33.0% lower MAE than DLinear on ETTh2. Against Transformer-based models, it achieves up to 13.7% lower MSE and 7.0% lower MAE than iTransformer on Weather, and up to 16.2% lower MSE and 16.8% lower MAE than PatchTST on Solar. It also reduces MAE by up to 15.9% over FilterTS on ETTm2, and achieves up to 17.1% lower MSE and 10.2% lower MAE than TimesNet on ETTh2.

These gains mainly come from abstracting effective temporal dynamics into L-Context, which explicitly encodes task-relevant variations as a dynamic intermediate context. This enables faster adaptation to data changes, mitigates response lag, and improves forecasting accuracy.

### 5.3. Ablation Study

We perform ablations to identify the sources of performance gains. For L-Context Generator, we ablate context usage and construction by removing the module (w/o L-Context), removing gating (w/o Gating), and replacing the L-context with randomly initialized learnable variables to disentangle structured context modeling from parameter increase (Rand-Context). For Struct-Aided Predictor, we ablate structural prior injection by removing intra-patch relative positional bases (w/o RelativePos) and replacing them with sequence-level positional representations to distinguish local relative positions from global absolute indices (GlobalPos). As shown in Table 2, the ablations degrade performance. Removing the L-Context Generator or gating reduces accuracy, indicating that structured context with gated modulation is essential for modeling key change segments; RandContext further underperforms, confirming that gains arise from context structure rather than added parameters. For the Struct-Aided Predictor, both w/o RelativePos and GlobalPos perform worse, showing that improvements stem from a patch-shared relative structural bias decoupled from global time indices, which promotes intra-segment pattern learning, reduces overfitting, and improves training stability.

### 5.4. Module Analysis

#### 5.4.1. PORTABILITY

To verify the portability of L-Context, we further conduct a transfer experiment in Table 3. We inject L-Context as an additional change cue into forecasting models of different paradigms, and evaluate it while keeping the main architecture of each baseline unchanged. We select three representative baselines, including the linear method DLinear, and the Transformer-based iTransformer and PatchTST. The results show that after introducing L-Context, the prediction errors of all three models decrease consistently. This indicates that the information carried by L-Context provides an intermediate representation that is more sensitive to dynamic changes and regime shifts, helping predictors with different structures adjust their inference for future evolution in a more timely manner, thereby improving prediction stability and overall accuracy. More importantly, the gains are consistent across heterogeneous backbones, suggesting that L-Context is not a trick tied to a specific architecture, but a model-agnostic signal for modeling changes.

#### 5.4.2. L-CONTEXT VISUALIZATION

To qualitatively inspect the contextual patterns learned by the L-Context Generator, we select a sample segment from the ECL dataset. We visualize the L-Context of the first 32 features and the first 32 time steps as a heatmap, and align it with a representative time-series segment for comparison (Figure 3). The heatmap shows a non-uniform and block-wise response pattern across both time and feature dimensions. Several high-response regions appear around intervals where the representative series exhibits visible local variations, such as rapid increases, slow declines, and rapid changes. In contrast, relatively smooth intervals tend to correspond to weaker responses in this example. This provides a qualitative view that the learned L-Context is not uniformly distributed over the input window, but presents stage-dependent response patterns that are visually associated with local temporal variations. This visualization provides an illustrative view of how L-Context exhibits non-uniform response patterns across time steps.

*Table 2.* Averaged ablation results. All ablated variants show degraded performance compared with the full L-Drive model.

| Datasets | Ours | | w/o L-Context | | w/o Gating | | RandContext | | w/o RelativePos | | GlobalPos | |
|---|---|---|---|---|---|---|---|---|---|---|---|---|
| | MSE | MAE | MSE | MAE | MSE | MAE | MSE | MAE | MSE | MAE | MSE | MAE |
| ETTm1 | **0.371** | **0.393** | 0.376 | 0.395 | 0.378 | 0.398 | 0.375 | 0.394 | 0.375 | 0.396 | 0.375 | 0.394 |
| ETTm2 | **0.273** | **0.320** | 0.274 | 0.322 | 0.274 | 0.321 | 0.273 | 0.322 | 0.277 | 0.324 | 0.273 | 0.321 |
| ETTh1 | **0.427** | 0.433 | 0.434 | 0.437 | 0.430 | **0.432** | 0.434 | 0.437 | 0.431 | 0.433 | 0.434 | **0.432** |
| ETTh2 | **0.367** | **0.396** | 0.375 | 0.401 | 0.372 | 0.398 | 0.382 | 0.402 | 0.374 | 0.400 | 0.375 | 0.398 |
| Weather | **0.235** | **0.267** | 0.243 | 0.272 | 0.238 | 0.268 | 0.241 | 0.272 | 0.240 | 0.270 | 0.237 | 0.269 |
| ECL | **0.169** | **0.266** | 0.177 | 0.267 | 0.176 | 0.271 | 0.178 | 0.268 | 0.178 | 0.273 | 0.174 | 0.272 |

*Table 3.* Averaged portability results. After introducing L-Context as an auxiliary signal, all baselines achieve performance improvements.

| Datasets | DLinear | | | | iTransformer | | | | PatchTST | | | |
|---|---|---|---|---|---|---|---|---|---|---|---|---|
| | Original | | +L-Context | | Original | | +L-Context | | Original | | +L-Context | |
| | MSE | MAE | MSE | MAE | MSE | MAE | MSE | MAE | MSE | MAE | MSE | MAE |
| ETTh1 | 0.456 | 0.452 | **0.451** | **0.438** | 0.463 | 0.454 | **0.460** | **0.451** | 0.469 | 0.455 | **0.447** | **0.442** |
| ETTm1 | 0.403 | 0.407 | **0.401** | **0.399** | 0.407 | 0.411 | **0.399** | **0.407** | 0.387 | 0.400 | **0.386** | **0.398** |
| Weather | **0.265** | 0.317 | **0.265** | **0.288** | 0.258 | 0.278 | **0.251** | **0.276** | 0.259 | 0.281 | **0.256** | **0.280** |
| ECL | 0.212 | 0.300 | **0.198** | **0.295** | 0.178 | **0.270** | **0.176** | 0.272 | 0.205 | 0.290 | **0.184** | **0.286** |

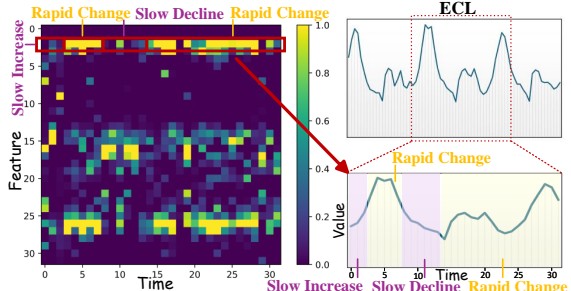

*Figure 3.* Visualization of L-Context on the ECL dataset.

*Table 4.* Average results on the lag error analysis.

| Datasets | L-Drive | | TimeBridge | | iTransformer | |
|---|---|---|---|---|---|---|
| | TailAUC | ExcessAUC | TailAUC | ExcessAUC | TailAUC | ExcessAUC |
| ETTh1 | **95.94** | **37.47** | 119.98 | 45.69 | 110.75 | 43.82 |
| EPF | **2.37** | **1.10** | 4.08 | 1.84 | 9.71 | 4.14 |
| Weather | **25.20** | **13.34** | 26.25 | 14.48 | 30.49 | 17.29 |
| Solar | **22.69** | **14.86** | 23.29 | 15.96 | 29.00 | 15.66 |

### 5.4.3. LAG ERROR ANALYSIS

To verify whether the lag-induced errors in change segments are mitigated, we detect change events on the ground-truth series by taking first-order differences and aggregating them across channels to obtain a change-intensity score. We select change times using the 90th-percentile threshold and enforce a minimum gap to avoid repeatedly counting the same change segment. We then accumulate absolute errors within the neighborhood window of each event to compute TailAUC. Further, we use the mean error over non-event regions as a steady-state baseline and accumulate only the part exceeding this baseline to compute ExcessAUC, which captures the extra error tail caused by changes. As shown

in Table 4 and the complete results in Appendix H, both TailAUC and ExcessAUC increase with the forecasting horizon for all three methods, indicating more pronounced error tails near switching points in long-horizon forecasting. However, L-Drive consistently achieves the lowest TailAUC and ExcessAUC across all datasets, suggesting faster post-change alignment and reduced excess error accumulation within the switching window.

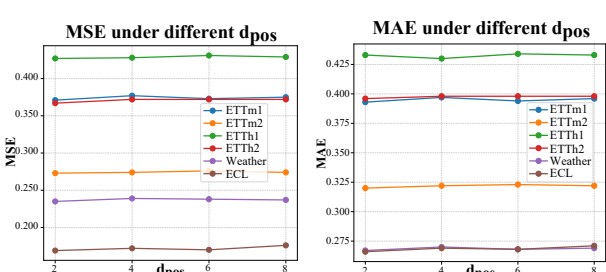

*Figure 4.* Averaged MSE and MAE results under different $d_{\text{pos}}$.

### 5.4.4. HYPERPARAMETER SENSITIVITY

To study the impact of the capacity of the patch-level relative position basis functions on model performance, we keep all other configurations unchanged and only vary the dimension of the relative position basis $d_{pos} \in \{2, 4, 6, 8\}$. We compare the results on six datasets. The experimental results are shown in Figure 4. Overall, the model demonstrates strong robustness to this hyperparameter. The errors on all datasets fluctuate only slightly as the dimension changes. It can be observed that smaller dimensions achieve the best or tied-best performance on most datasets. At the same time, increasing the dimension does not bring stable gains,

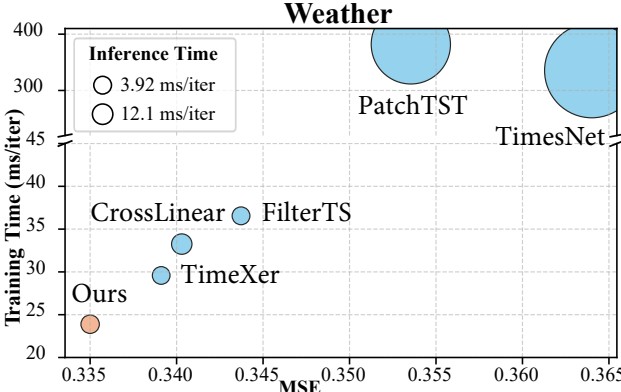

*Figure 5.* Computational efficiency analysis.

and slight degradation occurs in some scenarios. This indicates that the patch-level relative position is mainly used to distinguish relative relationships within a segment, and low-dimensional basis functions are sufficient to express these key structures. Higher-dimensional basis functions may introduce redundant representations, which can lead to slight overfitting.

### 5.4.5. COMPUTATIONAL EFFICIENCY

We conduct efficiency comparisons on the Weather dataset, strictly following the hyperparameter settings reported by each baseline. Under this setup, we plot the MSE, training time, and inference time of all models in a single bubble chart (Figure 5). Overall, our method achieves a better accuracy–efficiency trade-off. Specifically, Ours and CrossLinear both adopt the MLP paradigm and fall in the low-cost region, while Ours shifts further toward the lower-left, indicating lower error with even lower training cost. In contrast, Transformer-based models show clear cost divergence: TimeXer is relatively lightweight but still more expensive than Ours, whereas PatchTST incurs much higher training and inference costs. TimesNet also exhibits higher training and inference costs. FilterTS is faster at inference, but its error remains slightly higher than Ours. Overall, Ours is closest to the lower-left corner, demonstrating the best overall performance.

## 6. Conclusion

This paper focuses on the response-lag issue of the widely used Direct-Mapping paradigm in time-series forecasting, which often occurs under distribution shifts and trend turning points. To improve the model's sensitivity and adaptation to dynamic regime changes, we propose L-Drive, a change-aware forecasting framework. It uses L-Context to characterize time-evolving latent dynamics and to modulate increment representations, providing more timely change cues for prediction. We also introduce patch-shared relative

positional basis functions to enhance intra-segment structural modeling and reduce overfitting caused by absolute-position memorization. Extensive experiments demonstrate that our method is effective, efficient, and portable. Future work will explore finer-grained change modeling and improved robustness in broader real-world settings.

## Impact Statement

This paper presents work whose goal is to advance the field of Machine Learning. There are many potential societal consequences of our work, none which we feel must be specifically highlighted here.

## Acknowledgements

This work was supported in part by the following: the National Natural Science Foundation of China under Grant Nos. U24A20219, 62272281, U24A20328, the Yantai Natural Science Foundation under Grant No. 2024JCYJ034, and the Youth Innovation Technology Project of Higher School in Shandong Province under Grant No. 2023KJ212.

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

## A. Proof of Direct-Mapping Decomposition (Equation (2))

Consider a general form of a Direct-Mapping forecasting model:

$$\hat{y}_t = f_\theta(x_{1:t}), \tag{18}$$

where $x_{1:t}$ denotes the historical observations up to time $t$, and $f_\theta(\cdot)$ is a parameterized forecasting mapping. For simplicity, we denote the prediction output as $\hat{y}_t$. Since the influence of historical observations $x_{1:t-1}$ must be transmitted through a finite dimensional internal state, there must exist a state representation $s_{t-1}$ such that:

$$s_{t-1} = \phi_\theta(x_{1:t-1}), \tag{19}$$

here, $\phi_\theta(\cdot)$ denotes the process of summarizing historical information into a state. For analyzing prediction inertia, we consider the induced one-step recursion in the prediction space

$$\hat{y}_t = \psi_\theta(x_t, s_{t-1}), \tag{20}$$

here, $\psi_\theta(\cdot)$ denotes how the previous prediction affects the next prediction in the deployed recursive inference.

In general, the hidden state $s_{t-1}$ is not uniquely determined by the scalar output $\hat{y}_{t-1}$ for complex sequence models. To obtain a one-dimensional reduced-form recursion in the prediction space, we define

$$\psi_\theta(x_t, \hat{y}_{t-1}) \triangleq \mathbb{E}[\hat{y}_t \mid x_t, \hat{y}_{t-1}], \tag{21}$$

where the conditional expectation is taken under the data distribution induced by the deployed model. This mapping always exists (as the optimal $L_2$ predictor of $\hat{y}_t$ given $(x_t, \hat{y}_{t-1})$), and yields the reduced-form approximation

$$\hat{y}_t \approx \psi_\theta(x_t, \hat{y}_{t-1}) \tag{22}$$

in the sense of conditional mean.

We decompose the one-step forecasting operator into a part determined only by $x_t$ and a residual part introduced by $\hat{y}_{t-1}$. Specifically, we define:

$$\begin{aligned} g_\theta(x_t) &\triangleq \psi_\theta(x_t, 0), \\ h_{\theta,t}(\hat{y}) &\triangleq \psi_\theta(x_t, \hat{y}) - \psi_\theta(x_t, 0). \end{aligned} \tag{23}$$

Thus, we have an exact decomposition of the reduced-form operator:

$$\psi_\theta(x_t, \hat{y}_{t-1}) = g_\theta(x_t) + h_{\theta,t}(\hat{y}_{t-1}). \tag{24}$$

The approximation enters only when we linearize $h_{\theta,t}(\cdot)$ locally around an operating point $\bar{y}$ by omitting higher-order terms.

Assume that the reduced-form operator $\psi_\theta(x_t, \cdot)$ is locally differentiable in a neighborhood of $\bar{y}$.

We perform a first-order Taylor expansion of $h_{\theta,t}(\cdot)$ around a certain operating point $\bar{y}$:

$$\psi_\theta(x_t, \hat{y}_{t-1}) \approx g_\theta(x_t) + h_{\theta,t}(\bar{y}) + h'_{\theta,t}(\bar{y})(\hat{y}_{t-1} - \bar{y}), \tag{25}$$

here, $h'_{\theta,t}(\bar{y})$ represents the slope.

$$\begin{aligned} \psi_\theta(x_t, \hat{y}_{t-1}) &\approx g_\theta(x_t) + h_{\theta,t}(\bar{y}) + h'_{\theta,t}(\bar{y})\hat{y}_{t-1} - h'_{\theta,t}(\bar{y})\bar{y}, \\ \tilde{g}_{\theta,t}(x_t) &\triangleq g_\theta(x_t) + h_{\theta,t}(\bar{y}) - h'_{\theta,t}(\bar{y})\bar{y}, \\ \rho_t &\triangleq h'_{\theta,t}(\bar{y}), \\ \psi_\theta(x_t, \hat{y}_{t-1}) &\approx \rho_t \hat{y}_{t-1} + \tilde{g}_{\theta,t}(x_t). \end{aligned} \tag{26}$$

There exists a neighborhood $N(\bar{y})$ around the operating point $\bar{y}$ and a constant $L \in (0, 1)$ such that the one-step operator $\psi_\theta(x_t, \cdot)$ is locally contractive with respect to its second argument:

$$\left| \psi_\theta(x_t, u) - \psi_\theta(x_t, v) \right| \leq L \left| u - v \right|, \qquad \forall u, v \in \mathcal{N}(\bar{y}). \tag{27}$$

If $\psi_\theta(x_t, \cdot)$ is differentiable on $N(\bar{y})$, a sufficient condition for Equation (27) is

$$\sup_{u \in \mathcal{N}(\bar{y})} \left| \frac{\partial \psi_\theta(x_t, u)}{\partial u} \right| \leq L < 1. \tag{28}$$

Recall that Equation (23). Hence, $h_{\theta,t}(\hat{y}) \triangleq \psi_\theta(x_t, \hat{y}) - \psi_\theta(x_t, 0)$. Hence, $h'_{\theta,t}(\hat{y}) = \frac{\partial \psi_\theta(x_t, \hat{y})}{\partial \hat{y}}$, and therefore the slope $\rho_t$ in Equation (26) is

$$|\rho_t| = |h'_{\theta,t}(\bar{y})| = \left| \frac{\partial \psi_\theta(x_t, u)}{\partial u} \Big|_{u=\bar{y}} \right|. \tag{29}$$

Combining Equation (29) with Equation (28) yields

$$|\rho_t| \leq \sup_{u \in \mathcal{N}(\bar{y})} \left| \frac{\partial \psi_\theta(x_t, u)}{\partial u} \right| \leq L < 1, \tag{30}$$

which proves $|\rho_t| < 1$ in the operating neighborhood. Therefore the local autoregressive coefficient is generally time-varying ($\rho_t$ depends on $x_t$ through $\psi_\theta$). If $|\rho_t|$ is uniformly bounded by $L < 1$ in the operating region, one may use a representative constant $\rho$ (or $\rho_{\max} = L$) for concise analysis in the main text.

**Tightness of the reduced-form projection.**    We further discuss the reduced-form projection in Equation (21) is sufficiently tight for the subsequent local Taylor expansion and contraction argument to be meaningful.

The argument of Equation (21) is meaningful under the following three conditions, which are also supported by our empirical observations:

1. Small conditional projection residual. Define $\eta_t := \hat{y}_t - \psi_\theta(x_t, \hat{y}_{t-1})$. The tightness of the projection is governed by the magnitude of $\eta_t$. In our experiments, the full reduced-form proxy achieves consistently high $R^2$ (up to $\approx 0.99$) and low RMSE on held-out switching events, indicating that the conditional projection residual is small in the regions where the theory is applied.

2. Bounded local curvature. The Taylor expansion is performed within local switching neighborhoods, where the state deviation is limited. Under standard smoothness assumptions of neural networks on bounded domains, the second-order term is therefore controlled and remains a higher-order correction.

3. Sufficient contraction margin. The estimated coefficients satisfy $\rho \in (0, 1)$ across datasets and horizons, which implies the existence of a non-zero contraction margin $1 - \rho > 0$. As long as the approximation error is small relative to this margin, it only appears as a perturbation term in the bound and does not invalidate the contraction argument.

Taken together, these observations indicate that Equation (21) provides a sufficiently tight local approximation in practice, and thus the Taylor expansion and contraction-based lag bound in Section 4.3 are meaningful in the switching regimes of interest.

## B. Local Reduced-Form Fidelity Evaluation

To quantitatively evaluate how well Equation (2) approximates the local prediction behavior of single-forward-pass baselines, we collect successive predictions by sliding the lookback window, detect switching events from the ground-truth sequence, and construct local neighborhoods around each event following the protocol in Section 5.4.3. To avoid information leakage, we use event-level training and testing splits.

We compare three local proxy forms. The first is an inertia-only form:

$$\hat{y}_t \approx c + \rho \hat{y}_{t-1}, \tag{31}$$

which measures whether the current prediction can be explained by the previous prediction alone. The second is an update-only form:

$$\hat{y}_t \approx c + \alpha x_{\text{new}} + \beta x_{\text{old}}, \tag{32}$$

where $x_{\text{new}}$ and $x_{\text{old}}$ denote the newly added and removed inputs under the sliding-window mechanism, respectively. The third is the full reduced-form proxy:

$$\hat{y}_t \approx c + \rho\hat{y}_{t-1} + \alpha x_{\text{new}} + \beta x_{\text{old}}, \tag{33}$$

which combines prediction inertia with window-induced updates and serves as an empirical proxy of Equation (2).

We report $R^2$, RMSE, MAE, and the estimated inertia coefficient $\rho$. Here, $R^2$ measures the explained variance, where higher values indicate better approximation quality. RMSE and MAE measure fitting errors, where lower values indicate better fit. The coefficient $\rho$ quantifies the strength of local prediction inertia, while $\alpha$ and $\beta$ capture the contribution of newly added and removed inputs under the sliding-window setting.

The results in Tables 5 and 6 show that the inertia-only form explains a substantial portion of the local prediction behavior, confirming the presence of prediction inertia around switching events. However, it is consistently outperformed by the full form, indicating that inertia alone is insufficient. The update-only form is less stable and generally yields lower $R^2$ and larger fitting errors. In contrast, the full reduced-form proxy achieves the best overall performance by improving $R^2$ and reducing RMSE/MAE. These results indicate that the local prediction behavior of single-forward-pass baselines around switching events is better approximated by combining inertial dependence with window-induced updates.

## C. Increment-Related Interpretation of L-Context

This appendix provides further justification for interpreting L-Context as an increment-related representation. The first part gives a theoretical explanation: under local smoothness and bounded initial-state variation, the L-Context representation can be approximated by a function of the temporal difference sequence up to a bounded error. This shows that L-Context admits a difference-based interpretation. The second part provides empirical evidence by measuring the distance correlation between the raw difference signal and the learned L-Context representation on held-out data. Together, the theoretical bounded-error derivation and the empirical correlation analysis support the use of L-Context as an increment-related proxy in Equation (12).

### C.1. Bounded-Error Derivation

We show that, under local smoothness and bounded initial-state variation, the L-Context representation $z_t$ can be approximated by a function of the difference sequence $dx$ up to a bounded error, below we use $z_t$ and $dx$ to refer to both. By construction, the model defines a deterministic mapping

$$z_t = F(dx_1, \ldots, dx_t; x_0), \tag{34}$$

where $x_0$ denotes the initial state of the input sequence.

Since the sequence satisfies

$$x_t = x_0 + \sum_{i=1}^{t} dx_i, \tag{35}$$

the dependence of $z_t$ on the original inputs can be equivalently expressed through the difference sequence $(dx_1, \ldots, dx_t)$ together with the initial state $x_0$. Since the model is composed of smooth neural network components, the induced mapping $F$ is locally Lipschitz continuous on compact subsets of the input space. In particular, for any fixed difference sequence $dx$ and any two initial states $x_0, x_0' \in \mathcal{B}$, where $\mathcal{B}$ denotes a bounded set of admissible initial states in the dataset, there exists a constant $L > 0$ such that

$$\|F(dx; x_0) - F(dx; x_0')\| \leq L\|x_0 - x_0'\|. \tag{36}$$

This inequality shows that variations in the initial state induce only bounded changes in the output representation. Consequently, there exists a function $G$ depending only on $dx$ such that

$$\|z_t - G(dx)\| \leq L \cdot \text{diam}(\mathcal{B}), \tag{37}$$

which implies that the output representation can be approximated by a function of the difference sequence up to a bounded error. Therefore, the temporal difference sequence provides sufficient information to estimate the output representation.

*Table 5.* We evaluate three local proxy forms: (1) Inertia-only, $\hat{y}_t \approx c + \rho\hat{y}_{t-1}$; (2) Update-only, $\hat{y}_t \approx c + \alpha x_{\text{new}} + \beta x_{\text{old}}$; and (3) Full model, $\hat{y}_t \approx c + \rho\hat{y}_{t-1} + \alpha x_{\text{new}} + \beta x_{\text{old}}$, which combines both mechanisms and serves as a reduced-form proxy of Equation (2). Model performance is assessed using $R^2$ (variance explained; higher is better) and RMSE/MAE (absolute errors; lower is better). The coefficient $\rho$ quantifies the strength of prediction inertia, while $\alpha$ and $\beta$ capture the contribution of newly added and removed inputs under the sliding-window setting. Bold indicates the best results.

| Datasets | Pre_Lengths | Methods | TimeBridge | | | iTransformer | | |
|---|---|---|---|---|---|---|---|---|
| | | | $R^2$ | MAE | RMSE | $R^2$ | MAE | RMSE |
| ETTh1 | 96 | Inertia-only | 0.842 | 0.404 | 0.531 | 0.854 | 0.417 | 0.527 |
| | | Update-only | 0.842 | 0.413 | 0.531 | 0.873 | 0.407 | 0.491 |
| | | Full | **0.863** | **0.381** | **0.495** | **0.889** | **0.376** | **0.458** |
| | 192 | Inertia-only | 0.865 | 0.355 | 0.484 | 0.862 | 0.401 | 0.506 |
| | | Update-only | 0.852 | 0.394 | 0.508 | 0.871 | 0.392 | 0.489 |
| | | Full | **0.881** | **0.343** | **0.456** | **0.886** | **0.368** | **0.459** |
| | 336 | Inertia-only | 0.872 | 0.322 | 0.443 | 0.883 | 0.371 | 0.469 |
| | | Update-only | 0.853 | 0.375 | 0.475 | 0.848 | 0.437 | 0.535 |
| | | Full | **0.889** | **0.318** | **0.415** | **0.894** | **0.365** | **0.448** |
| | 720 | Inertia-only | 0.906 | 0.245 | 0.306 | 0.888 | 0.311 | 0.387 |
| | | Update-only | 0.845 | 0.315 | 0.394 | 0.891 | 0.294 | 0.382 |
| | | Full | **0.917** | **0.235** | **0.288** | **0.910** | **0.274** | **0.345** |
| ETTh2 | 96 | Inertia-only | 0.936 | 0.136 | 0.177 | 0.940 | 0.122 | 0.164 |
| | | Update-only | 0.889 | 0.182 | 0.233 | 0.796 | 0.238 | 0.303 |
| | | Full | **0.967** | **0.101** | **0.127** | **0.967** | **0.093** | **0.122** |
| | 192 | Inertia-only | 0.958 | 0.105 | 0.135 | 0.953 | 0.109 | 0.145 |
| | | Update-only | 0.846 | 0.203 | 0.259 | 0.806 | 0.234 | 0.295 |
| | | Full | **0.975** | **0.081** | **0.104** | **0.968** | **0.093** | **0.120** |
| | 336 | Inertia-only | 0.984 | 0.069 | 0.086 | 0.965 | 0.099 | 0.132 |
| | | Update-only | 0.766 | 0.254 | 0.330 | 0.768 | 0.269 | 0.343 |
| | | Full | **0.990** | **0.055** | **0.068** | **0.978** | **0.082** | **0.107** |
| | 720 | Inertia-only | **0.988** | **0.061** | **0.079** | 0.975 | 0.086 | 0.116 |
| | | Update-only | 0.539 | 0.367 | 0.483 | 0.726 | 0.302 | 0.382 |
| | | Full | 0.987 | 0.063 | 0.081 | **0.981** | **0.076** | **0.100** |
| ETTm1 | 96 | Inertia-only | 0.969 | 0.154 | 0.240 | 0.978 | 0.134 | 0.197 |
| | | Update-only | 0.981 | 0.136 | 0.190 | 0.952 | 0.210 | 0.287 |
| | | Full | **0.990** | **0.097** | **0.138** | **0.985** | **0.119** | **0.160** |
| | 192 | Inertia-only | 0.970 | 0.138 | 0.211 | 0.978 | 0.125 | 0.181 |
| | | Update-only | 0.980 | 0.123 | 0.172 | 0.958 | 0.183 | 0.252 |
| | | Full | **0.990** | **0.084** | **0.121** | **0.988** | **0.101** | **0.137** |
| | 336 | Inertia-only | 0.972 | 0.142 | 0.217 | 0.981 | 0.119 | 0.173 |
| | | Update-only | 0.975 | 0.154 | 0.205 | 0.955 | 0.190 | 0.268 |
| | | Full | **0.986** | **0.109** | **0.151** | **0.988** | **0.100** | **0.139** |
| | 720 | Inertia-only | 0.971 | 0.132 | 0.198 | 0.981 | 0.113 | 0.159 |
| | | Update-only | 0.969 | 0.153 | 0.206 | 0.919 | 0.240 | 0.330 |
| | | Full | **0.983** | **0.108** | **0.154** | **0.984** | **0.109** | **0.146** |
| ETTm2 | 96 | Inertia-only | 0.978 | 0.089 | 0.117 | 0.986 | 0.067 | 0.088 |
| | | Update-only | 0.979 | 0.091 | 0.114 | 0.934 | 0.147 | 0.191 |
| | | Full | **0.994** | **0.049** | **0.063** | **0.992** | **0.052** | **0.068** |
| | 192 | Inertia-only | 0.974 | 0.097 | 0.131 | 0.990 | 0.054 | 0.074 |
| | | Update-only | 0.966 | 0.115 | 0.149 | 0.883 | 0.190 | 0.248 |
| | | Full | **0.991** | **0.058** | **0.076** | **0.993** | **0.046** | **0.061** |
| | 336 | Inertia-only | 0.978 | 0.084 | 0.113 | 0.993 | 0.046 | 0.062 |
| | | Update-only | 0.957 | 0.118 | 0.159 | 0.888 | 0.183 | 0.243 |
| | | Full | **0.992** | **0.053** | **0.070** | **0.996** | **0.035** | **0.048** |
| | 720 | Inertia-only | 0.983 | 0.071 | 0.095 | 0.994 | 0.039 | 0.053 |
| | | Update-only | 0.937 | 0.146 | 0.186 | 0.871 | 0.195 | 0.255 |
| | | Full | **0.991** | **0.053** | **0.069** | **0.997** | **0.029** | **0.041** |

*Table 6.* Estimated inertia coefficient $\rho$ from the local proxy model for TimeBridge and iTransformer across datasets and forecasting horizons. The coefficient $\rho$ quantifies the strength of prediction inertia, with larger values indicating that current predictions are more strongly tied to previous predictions.

| Datasets | Pre_Lengths | TimeBridge $\rho$ | iTransformer $\rho$ | Datasets | Pre_Lengths | TimeBridge $\rho$ | iTransformer $\rho$ |
|---|---|---|---|---|---|---|---|
| **ETTh1** | **96** | 0.409 | 0.340 | **ETTh2** | **96** | 0.606 | 0.722 |
| | **192** | 0.569 | 0.409 | | **192** | 0.734 | 0.760 |
| | **336** | 0.599 | 0.639 | | **336** | 0.857 | 0.821 |
| | **720** | 0.725 | 0.520 | | **720** | 0.907 | 0.836 |
| **ETTm1** | **96** | 0.391 | 0.678 | **ETTm2** | **96** | 0.507 | 0.746 |
| | **192** | 0.423 | 0.665 | | **192** | 0.559 | 0.841 |
| | **336** | 0.514 | 0.719 | | **336** | 0.602 | 0.857 |
| | **720** | 0.583 | 0.802 | | **720** | 0.732 | 0.895 |

## C.2. Distance-Correlation Analysis

We compute the distance correlation (dCor) between the raw differenced signal and the L-Context representations on multiple real-world datasets. Specifically, we calculate the pairwise distances between time steps and measure the correlation between the two distance structures. A value closer to 1 indicates a stronger correlation. The focus is on whether the internal variation structure in the differenced sequence is preserved in the L-Context representation space. The results are shown in Table 7. The correlation values across all datasets remain at a high level, and the standard deviations across all tasks are very small, indicating that there exists a strong and stable structural correlation between the raw differenced signal and L-Context in general. This suggests that L-Context indeed encodes information related to variation patterns.

*Table 7.* dCor results across datasets and prediction lengths.

| Datasets | Pre_Lengths/Subsets | Mean | Std | Datasets | Pre_Lengths/Subsets | Mean | Std |
|---|---|---|---|---|---|---|---|
| **ETTh1** | **96** | 0.845 | 0.003 | **ETTh2** | **96** | 0.812 | 0.003 |
| | **192** | 0.832 | 0.003 | | **192** | 0.820 | 0.002 |
| | **336** | 0.840 | 0.003 | | **336** | 0.796 | 0.004 |
| | **720** | 0.834 | 0.005 | | **720** | 0.795 | 0.005 |
| | **Avg** | 0.838 | 0.003 | | **Avg** | 0.806 | 0.003 |
| **ETTm1** | **96** | 0.805 | 0.007 | **ETTm2** | **96** | 0.817 | 0.005 |
| | **192** | 0.798 | 0.004 | | **192** | 0.743 | 0.004 |
| | **336** | 0.796 | 0.004 | | **336** | 0.840 | 0.003 |
| | **720** | 0.803 | 0.010 | | **720** | 0.778 | 0.008 |
| | **Avg** | 0.800 | 0.006 | | **Avg** | 0.794 | 0.005 |
| **Weather** | **96** | 0.772 | 0.007 | **Solar** | **96** | 0.889 | 0.002 |
| | **192** | 0.759 | 0.004 | | **192** | 0.849 | 0.009 |
| | **336** | 0.796 | 0.007 | | **336** | 0.952 | 0.003 |
| | **720** | 0.800 | 0.007 | | **720** | 0.829 | 0.003 |
| | **Avg** | 0.782 | 0.006 | | **Avg** | 0.880 | 0.004 |
| **EPF** | **NP** | 0.847 | 0.001 | **EPF** | **PJM** | 0.932 | 0.000 |
| | **BE** | 0.925 | 0.001 | | **FR** | 0.854 | 0.003 |
| | **DE** | 0.900 | 0.004 | | **Avg** | 0.892 | 0.002 |

## D. Proof of L-Context Lag Mitigation (Equations (16) and (17))

For the common monotone-retention case we have $\rho \in (0,1)$; the following bound is stated in the more general form $|\rho| < 1$.

Substituting Equation (14) into Equation (13) and simplifying using $\Delta g_t = g_t - g_{t-1}$, we obtain:

$$
\begin{aligned}
\hat{y}_t^L &= \rho \hat{y}_{t-1}^L + (1-\rho)g_t + \rho(\Delta g_t + \varepsilon_t) \\
&= \rho \hat{y}_{t-1}^L + (1-\rho)g_t + \rho(g_t - g_{t-1}) + \rho\varepsilon_t \\
&= \rho \hat{y}_{t-1}^L + g_t - \rho g_{t-1} + \rho\varepsilon_t.
\end{aligned}
\tag{38}
$$

Subtracting $g_t$ from both sides and using $e_{t-1} = \hat{y}_{t-1}^L - g_{t-1}$, we obtain:

$$
\begin{aligned}
\hat{y}_t^L - g_t &= \rho(\hat{y}_{t-1}^L - g_{t-1}) + \rho\varepsilon_t, \\
e_t &= \rho e_{t-1} + \rho\varepsilon_t.
\end{aligned}
\tag{39}
$$

For any $K \geqslant 1$,

$$
e_t = \rho^K e_{t-K} + \rho \sum_{k=0}^{K-1} \rho^k \varepsilon_{t-k}.
\tag{40}
$$

This identity holds for any constant $\rho$; the subsequent bound is stated using $|\rho|$.

Therefore,

$$
|e_t| \leq |\rho|^K |e_{t-K}| + |\rho| \sum_{k=0}^{K-1} |\rho|^k |\varepsilon_{t-k}|.
\tag{41}
$$

If $|\varepsilon_t| \leq \bar{\varepsilon}$, then

$$
\limsup_{t \to \infty} |e_t| \leq \frac{|\rho|}{1-|\rho|} \bar{\varepsilon}.
\tag{42}
$$

Proof completed.

**Extension to time-varying retention.** As a worst-case generalization, we allow the retention coefficient in the error recursion to vary over time:

$$
e_t = \rho_t e_{t-1} + \rho_t \varepsilon_t, \qquad |\rho_t| \leq \rho_{\max} < 1,
\tag{43}
$$

then

$$
|e_t| \leq \rho_{\max} |e_{t-1}| + \rho_{\max} |\varepsilon_t|.
\tag{44}
$$

Iterating the inequality yields

$$
\limsup_{t \to \infty} |e_t| \leq \frac{\rho_{\max}}{1-\rho_{\max}} \bar{\varepsilon},
\tag{45}
$$

which reduces to the constant-$\rho$ bound when $\rho_{\max} = |\rho|$.

## E. Dataset Descriptions

ETT (ETTh1, ETTh2, ETTm1, ETTm2): ETT is a classic benchmark family for "electricity transformer temperature" forecasting. Among them, ETTh1/ETTh2 are sampled hourly, while ETTm1/ETTm2 are sampled every 15 minutes. The datasets are widely used for long-horizon multi-step time-series forecasting benchmarks.

Weather: This dataset contains 21 meteorological variables (e.g., temperature, humidity, wind, etc.) and is sampled every 10 minutes. It is often used to evaluate a model's ability to handle multivariate series where strong seasonality and short-term fluctuations coexist.

Solar_Energy: The Solar-Energy dataset typically refers to a collection of PV power generation time series from solar plants, sampled every 10 minutes. It is commonly used to test forecasting stability and generalization in renewable-energy scenarios, which are strongly weather-driven and highly volatile.

*Table 8.* Dataset descriptions.

| Datasets | Timesteps | Features | Frequency | Information |
|----------|-----------|----------|-----------|-------------|
| **ETTh1** | 17,420 | 7 | 1 hour | Electricity |
| **ETTh2** | 17,420 | 7 | 1 hour | Electricity |
| **ETTm1** | 69,680 | 7 | 15 mins | Electricity |
| **ETTm2** | 69,680 | 7 | 15 mins | Electricity |
| **Weather** | 52,696 | 21 | 10 mins | Weather |
| **Solar_Energy** | 52,560 | 137 | 10 mins | Energy |
| **Electricity** | 26,304 | 321 | 1 hour | Electricity |

Electricity (ECL): ECL is a classic electricity load forecasting dataset that records electricity consumption curves from a large number of users/meters over time, making it a typical multivariate (or multi-series) time-series dataset.

The detailed dataset descriptions are shown in Table 8.

## F. Detailed Experimental Setting

All experiments are conducted under the same hardware environment using an NVIDIA GeForce RTX 3090 with 24 GB GPU memory. The batch size is set in the range of [4, 32], and the learning rate is searched within [1e-5, 1e-3]. Following the settings in TFB (Qiu et al., 2024) and TAB (Qiu et al., 2025b), we do not apply the "Drop Last" trick to ensure a fair comparison. For long-term time series forecasting, all baselines use the same input length of 96, and all other hyperparameters follow the official implementations and the default configurations reported in the original papers; when an explicit configuration script is unavailable, we use the default parameter settings in the released codebase. The complete tables for the main forecasting task and the ablation study are provided in Tables 9, 11 and 12. We also report the MAPE forecasting results in Table 10.

## G. Portability Evaluation Setup

To assess the portability of L-Context across different forecasting backbones, we inject L-Context as an auxiliary change cue into representative Direct-Mapping baselines while keeping their main architectures unchanged. We consider three heterogeneous models, including DLinear, iTransformer, and PatchTST. For each baseline, we compute the L-Context and fuse it with the original input through a simple weighted combination at the input stage. The fused sequence is then fed into the baseline model in the standard way, without modifying its encoder/decoder blocks or prediction head.

All experiments follow the same data preprocessing and evaluation protocol as the main long-term forecasting setting. We keep the input length and forecasting horizons consistent with the main experiments, and strictly follow the official hyperparameter settings and training pipelines of each baseline. The only additional component is the fusion operation used to combine L-Context with the original input, which does not change the overall model structure. We report MSE/MAE and compare each baseline with and without L-Context injection to quantify the effect of L-Context on different backbones. The complete tables are provided in Table 13.

## H. Lag Error Analysis Evaluation Setup

To quantify prediction lag and error tails around change segments in Tables 4 and 14, we introduce an event-aligned evaluation based on TailAUC and ExcessAUC. We first obtain a continuous prediction trajectory on the test set by merging the rolling-window forecasts so that each time step is associated with a single prediction. We then compute the per-time-step absolute error sequence between predictions and ground truth.

Change events are automatically detected from the ground-truth series. Specifically, we compute first-order differences of the ground truth and aggregate them across channels to obtain a change-intensity score for each time step. We select event times whose scores exceed the 90th-percentile threshold, and enforce a minimum temporal gap between consecutive events to avoid repeatedly counting the same change segment.

*Table 9.* Full results on the forecasting task. **Bold** indicates the best results, and underlining indicates the second-best results.

| Datasets | Methods | L-Drive (Ours) | | TimeBridge (2025) | | CrossLinear (2025) | | FilterTS (2025) | | PatchMLP (2025) | | iTransformer (2024) | | TimeXer (2024) | | PatchTST (2023) | | TimesNet (2023) | | DLinear (2023) | |
|---|---|---|---|---|---|---|---|---|---|---|---|---|---|---|---|---|---|---|---|---|---|
| | Metrics | MSE | MAE | MSE | MAE | MSE | MAE | MSE | MAE | MSE | MAE | MSE | MAE | MSE | MAE | MSE | MAE | MSE | MAE | MSE | MAE |
| ETTh1 | 96 | **0.368** | 0.393 | 0.375 | **0.390** | 0.374 | 0.393 | 0.374 | 0.391 | 0.391 | 0.403 | 0.394 | 0.409 | 0.382 | 0.403 | 0.414 | 0.419 | 0.384 | 0.402 | 0.386 | 0.400 |
| | 192 | **0.417** | 0.426 | 0.428 | 0.421 | 0.422 | 0.424 | 0.424 | **0.421** | 0.444 | 0.431 | 0.448 | 0.441 | 0.429 | 0.435 | 0.460 | 0.445 | 0.436 | 0.429 | 0.437 | 0.432 |
| | 336 | **0.455** | 0.446 | 0.470 | 0.443 | 0.459 | 0.447 | 0.464 | **0.441** | 0.490 | 0.456 | 0.491 | 0.464 | 0.468 | 0.448 | 0.501 | 0.466 | 0.491 | 0.469 | 0.481 | 0.459 |
| | 720 | **0.455** | **0.457** | 0.481 | 0.481 | 0.467 | 0.465 | 0.470 | 0.466 | 0.528 | 0.496 | 0.519 | 0.502 | 0.469 | 0.461 | 0.500 | 0.488 | 0.521 | 0.500 | 0.519 | 0.516 |
| | Avg | **0.424** | **0.430** | 0.438 | 0.434 | 0.431 | 0.432 | 0.433 | **0.430** | 0.463 | 0.447 | 0.463 | 0.454 | 0.437 | 0.437 | 0.469 | 0.455 | 0.458 | 0.450 | 0.456 | 0.452 |
| ETTh2 | 96 | **0.282** | 0.336 | 0.286 | **0.333** | 0.282 | 0.337 | 0.290 | 0.338 | 0.305 | 0.353 | 0.297 | 0.349 | 0.286 | 0.338 | 0.302 | 0.348 | 0.340 | 0.374 | 0.333 | 0.387 |
| | 192 | **0.358** | 0.386 | 0.364 | **0.385** | 0.360 | 0.387 | 0.374 | 0.390 | 0.402 | 0.410 | 0.380 | 0.400 | 0.363 | 0.389 | 0.388 | 0.400 | 0.402 | 0.414 | 0.477 | 0.476 |
| | 336 | **0.403** | 0.421 | 0.416 | 0.425 | 0.405 | 0.422 | 0.406 | **0.420** | 0.436 | 0.437 | 0.428 | 0.432 | 0.414 | 0.423 | 0.426 | 0.433 | 0.452 | 0.452 | 0.594 | 0.541 |
| | 720 | 0.421 | 0.439 | 0.435 | 0.448 | 0.424 | 0.439 | 0.418 | 0.437 | 0.437 | 0.449 | 0.427 | 0.445 | **0.408** | **0.432** | 0.431 | 0.446 | 0.462 | 0.468 | 0.831 | 0.657 |
| | Avg | **0.366** | **0.396** | 0.375 | 0.397 | 0.368 | 0.396 | 0.372 | 0.396 | 0.395 | 0.412 | 0.383 | 0.407 | 0.368 | 0.396 | 0.387 | 0.407 | 0.414 | 0.427 | 0.559 | 0.515 |
| ETTm1 | 96 | **0.310** | 0.354 | 0.312 | **0.344** | 0.311 | 0.354 | 0.321 | 0.360 | 0.324 | 0.362 | 0.341 | 0.376 | 0.318 | 0.356 | 0.329 | 0.367 | 0.338 | 0.375 | 0.345 | 0.372 |
| | 192 | 0.354 | 0.379 | 0.366 | **0.377** | 0.352 | 0.379 | 0.363 | 0.382 | 0.369 | 0.385 | 0.381 | 0.395 | 0.362 | 0.383 | 0.367 | 0.385 | 0.374 | 0.387 | 0.380 | 0.389 |
| | 336 | **0.381** | 0.402 | 0.403 | 0.404 | **0.381** | **0.401** | 0.395 | 0.403 | 0.402 | 0.407 | 0.417 | 0.418 | 0.395 | 0.407 | 0.399 | 0.410 | 0.410 | 0.411 | 0.413 | 0.413 |
| | 720 | **0.437** | **0.437** | 0.454 | 0.438 | 0.439 | 0.439 | 0.462 | 0.438 | 0.474 | 0.446 | 0.487 | 0.456 | 0.452 | 0.441 | 0.454 | 0.439 | 0.478 | 0.450 | 0.474 | 0.453 |
| | Avg | **0.371** | 0.393 | 0.383 | **0.391** | **0.371** | 0.393 | 0.385 | 0.396 | 0.392 | 0.400 | 0.407 | 0.411 | 0.382 | 0.397 | 0.387 | 0.400 | 0.400 | 0.406 | 0.403 | 0.407 |
| ETTm2 | 96 | **0.170** | 0.253 | 0.175 | **0.252** | **0.170** | 0.254 | 0.172 | 0.255 | 0.180 | 0.263 | 0.184 | 0.270 | 0.171 | 0.256 | 0.175 | 0.259 | 0.187 | 0.267 | 0.193 | 0.292 |
| | 192 | **0.236** | **0.298** | 0.246 | 0.299 | **0.236** | **0.298** | 0.237 | 0.299 | 0.244 | 0.306 | 0.251 | 0.312 | 0.237 | 0.299 | 0.241 | 0.302 | 0.249 | 0.309 | 0.284 | 0.362 |
| | 336 | **0.293** | **0.335** | 0.311 | 0.341 | 0.294 | 0.336 | 0.299 | 0.398 | 0.312 | 0.349 | 0.314 | 0.350 | 0.296 | 0.338 | 0.305 | 0.343 | 0.321 | 0.351 | 0.369 | 0.427 |
| | 720 | 0.392 | 0.394 | 0.403 | 0.397 | **0.388** | 0.394 | 0.397 | 0.394 | 0.411 | 0.407 | 0.411 | 0.405 | 0.392 | 0.394 | 0.402 | 0.400 | 0.408 | 0.403 | 0.554 | 0.522 |
| | Avg | 0.273 | **0.320** | 0.284 | 0.322 | **0.272** | 0.321 | 0.276 | 0.337 | 0.287 | 0.331 | 0.290 | 0.334 | 0.274 | 0.322 | 0.281 | 0.326 | 0.291 | 0.333 | 0.350 | 0.401 |
| ECL | 96 | 0.141 | 0.240 | **0.138** | **0.232** | 0.139 | 0.237 | 0.151 | 0.245 | 0.160 | 0.257 | 0.148 | 0.240 | 0.140 | 0.242 | 0.181 | 0.270 | 0.168 | 0.272 | 0.197 | 0.282 |
| | 192 | **0.156** | 0.253 | 0.157 | **0.251** | 0.157 | 0.254 | 0.163 | 0.256 | 0.176 | 0.271 | 0.162 | 0.253 | 0.157 | 0.256 | 0.188 | 0.274 | 0.184 | 0.289 | 0.196 | 0.285 |
| | 336 | **0.172** | 0.272 | 0.178 | 0.272 | 0.176 | 0.275 | 0.180 | 0.274 | 0.197 | 0.292 | 0.178 | **0.269** | 0.176 | 0.275 | 0.204 | 0.293 | 0.198 | 0.300 | 0.209 | 0.301 |
| | 720 | **0.206** | **0.298** | 0.226 | 0.312 | 0.221 | 0.315 | 0.224 | 0.311 | 0.249 | 0.333 | 0.225 | 0.317 | 0.211 | 0.306 | 0.246 | 0.324 | 0.220 | 0.320 | 0.245 | 0.333 |
| | Avg | **0.169** | **0.266** | 0.175 | 0.267 | 0.173 | 0.270 | 0.180 | 0.272 | 0.196 | 0.288 | 0.178 | 0.270 | 0.171 | 0.270 | 0.205 | 0.290 | 0.193 | 0.295 | 0.212 | 0.300 |
| Weather | 96 | **0.150** | **0.199** | 0.169 | 0.206 | 0.154 | 0.202 | 0.162 | 0.207 | 0.164 | 0.210 | 0.174 | 0.214 | 0.157 | 0.205 | 0.177 | 0.218 | 0.172 | 0.220 | 0.196 | 0.255 |
| | 192 | **0.199** | **0.244** | 0.219 | 0.249 | 0.200 | 0.246 | 0.209 | 0.252 | 0.211 | 0.251 | 0.221 | 0.254 | 0.204 | 0.247 | 0.225 | 0.259 | 0.219 | 0.261 | 0.237 | 0.296 |
| | 336 | **0.256** | 0.286 | 0.274 | 0.290 | 0.257 | 0.286 | 0.263 | 0.292 | 0.269 | 0.294 | 0.278 | 0.296 | 0.261 | 0.290 | 0.278 | 0.297 | 0.280 | 0.306 | 0.283 | 0.335 |
| | 720 | **0.335** | **0.338** | 0.353 | 0.341 | 0.340 | 0.343 | 0.344 | 0.344 | 0.349 | 0.345 | 0.358 | 0.349 | 0.340 | 0.341 | 0.354 | 0.348 | 0.365 | 0.359 | 0.345 | 0.381 |
| | Avg | **0.235** | **0.267** | 0.254 | 0.271 | 0.238 | 0.269 | 0.245 | 0.274 | 0.248 | 0.275 | 0.258 | 0.278 | 0.241 | 0.271 | 0.259 | 0.281 | 0.259 | 0.287 | 0.265 | 0.317 |
| Solar | 96 | 0.196 | 0.238 | **0.192** | **0.207** | 0.194 | 0.236 | 0.214 | 0.260 | 0.229 | 0.265 | 0.203 | 0.237 | 0.210 | 0.261 | 0.234 | 0.286 | 0.250 | 0.292 | 0.290 | 0.378 |
| | 192 | 0.230 | 0.261 | **0.223** | **0.231** | 0.225 | 0.258 | 0.247 | 0.284 | 0.270 | 0.287 | 0.233 | 0.261 | 0.236 | 0.273 | 0.267 | 0.310 | 0.296 | 0.318 | 0.320 | 0.398 |
| | 336 | 0.245 | 0.273 | **0.243** | **0.250** | 0.246 | 0.274 | 0.269 | 0.298 | 0.302 | 0.307 | 0.248 | 0.273 | 0.247 | 0.282 | 0.290 | 0.315 | 0.319 | 0.330 | 0.353 | 0.415 |
| | 720 | 0.248 | 0.276 | 0.250 | **0.257** | **0.245** | 0.274 | 0.264 | 0.292 | 0.308 | 0.311 | 0.249 | 0.275 | 0.250 | 0.286 | 0.289 | 0.317 | 0.338 | 0.337 | 0.356 | 0.413 |
| | Avg | 0.230 | 0.262 | **0.227** | **0.236** | 0.228 | 0.260 | 0.248 | 0.283 | 0.277 | 0.293 | 0.233 | 0.262 | 0.236 | 0.275 | 0.270 | 0.307 | 0.301 | 0.319 | 0.330 | 0.401 |
| EPF | NP | **0.288** | **0.309** | 0.379 | 0.382 | 0.289 | 0.312 | 0.300 | 0.316 | 0.313 | 0.332 | 0.360 | 0.358 | 0.301 | 0.324 | 0.298 | 0.329 | 0.330 | 0.349 | 0.340 | 0.369 |
| | PJM | **0.072** | **0.169** | 0.167 | 0.284 | 0.073 | 0.170 | 0.089 | 0.196 | 0.094 | 0.200 | 0.080 | 0.182 | 0.078 | 0.182 | 0.086 | 0.194 | 0.079 | 0.178 | 0.126 | 0.246 |
| | BE | **0.141** | 0.167 | 0.151 | 0.169 | 0.145 | 0.168 | 0.179 | 0.210 | 0.163 | 0.189 | 0.145 | 0.172 | 0.148 | 0.168 | 0.155 | 0.176 | 0.151 | 0.176 | 0.176 | 0.236 |
| | FR | 0.154 | 0.153 | **0.147** | **0.150** | 0.158 | 0.161 | 0.175 | 0.187 | 0.183 | 0.182 | 0.151 | 0.158 | 0.153 | 0.159 | 0.178 | 0.192 | 0.157 | 0.164 | 0.189 | 0.221 |
| | DE | **0.183** | **0.238** | 0.203 | 0.241 | 0.193 | 0.241 | 0.204 | 0.256 | 0.201 | 0.258 | 0.211 | 0.264 | 0.193 | 0.246 | 0.198 | 0.253 | 0.201 | 0.258 | 0.235 | 0.304 |
| | Avg | **0.168** | **0.207** | 0.209 | 0.245 | 0.172 | 0.210 | 0.190 | 0.233 | 0.191 | 0.232 | 0.189 | 0.227 | 0.175 | 0.216 | 0.183 | 0.229 | 0.184 | 0.225 | 0.218 | 0.275 |

For each detected event, we evaluate a post-event neighborhood window of fixed length. TailAUC is computed by accumulating absolute errors within the event window, measuring the overall magnitude of the error tail after a change. To further isolate the excess error induced by changes, we estimate a steady-state baseline as the mean absolute error over all non-event time points, and compute ExcessAUC by accumulating only the part of the window errors that exceeds this baseline.

## I. Hyperparameter Sensitivity Evaluation Setup

We conduct parameter sensitivity experiments to study how key hyperparameters affect the performance of L-Drive. Firstly, we vary the dimensionality of the relative positional basis functions while keeping all other model components and training configurations unchanged. We evaluate multiple settings ($d_{pos} \in \{2, 4, 6, 8\}$) under the same data preprocessing, input length, forecasting horizons, and evaluation protocol as the main experiments. Secondly, we vary the Conv1D kernel size while keeping all other model components and training configurations unchanged. We evaluate multiple settings ($KernelSize \in \{2, 3, 4, 6\}$) under the same data preprocessing, input length, forecasting horizons, and evaluation protocol as the main experiments.

This controlled protocol ensures that the observed performance differences can be attributed to the target hyperparameter,

*Table 10.* MAPE results on the forecasting task. **Bold** indicates the best results.

| Datasets | Methods | L-Drive | CrossLinear | TimeBridge | Datasets | Methods | L-Drive | CrossLinear | TimeBridge |
|---|---|---|---|---|---|---|---|---|---|
| **ETTh1** | **96** | **9.537** | 9.571 | 9.592 | **ETTm2** | **96** | 1.099 | **1.098** | 1.111 |
| | **192** | 9.859 | 9.869 | **9.739** | | **192** | 1.232 | **1.221** | 1.267 |
| | **336** | **9.447** | 10.081 | 9.642 | | **336** | 1.346 | **1.342** | 1.378 |
| | **720** | **9.287** | 9.920 | 9.734 | | **720** | **1.535** | 1.577 | 1.588 |
| | **Avg** | **9.533** | 9.860 | 9.677 | | **Avg** | **1.303** | 1.309 | 1.336 |
| **Solar** | **96** | **1.932** | 1.960 | 1.937 | **EPF** | **NP** | **1.443** | 1.496 | 1.847 |
| | **192** | **2.068** | 2.132 | 2.076 | | **PJM** | **1.700** | 1.715 | 2.981 |
| | **336** | 2.131 | 2.177 | **2.124** | | **BE** | **1.357** | 1.678 | 1.721 |
| | **720** | **2.201** | 2.262 | 2.202 | | **FR** | **1.250** | 1.371 | 1.407 |
| | **Avg** | **2.083** | 2.133 | 2.085 | | **DE** | 4.180 | **4.173** | 5.398 |
| | | | | | | **Avg** | **1.986** | 2.087 | 2.671 |

rather than confounding changes in model capacity or training conditions. The complete table is provided in Tables 15 and 16.

## J. Computational Efficiency Evaluation Setup

We conduct efficiency comparisons on the Weather dataset under a unified hardware environment using an NVIDIA GeForce RTX 3090 (24GB). For a fair comparison, all baselines use the same input length of 96 for long-term time series forecasting, while all other hyperparameters and training details strictly follow the official implementations and default configurations reported in the original papers. The complete table is provided in Table 17.

## K. Analysis of the gating mechanism (Equation (6))

To further examine whether the proposed gate selectively modulates difference components rather than simply re-weighting them uniformly, we conduct an event-conditioned analysis at test time. The intuition is that meaningful temporal variations are more likely to occur around event regions, whereas increments in non-event regions are more likely to reflect transient high-frequency fluctuations introduced by differencing. Therefore, a noise-aware gating mechanism is expected to retain a larger fraction of differential information in event regions while suppressing less informative increments in relatively stable regions.

Specifically, using externally defined event windows (following the same event detection procedure as in Sec. 5.4.3), we analyze all feature channels during a forward pass at test time. Since this analysis focuses on the immediate local response of the model, and high-frequency noise introduced by differencing is typically transient and short-lived, we adopt a short window of size 3 to isolate short-term dynamics and avoid contamination from longer-term effects. Within these event and non-event regions, we compare (i) the average gate value $E[m \mid event]$ vs. $E[m \mid non\text{-}event]$, and (ii) the retained increment ratio $\frac{|\Delta x \cdot \text{gate}|}{|\Delta x| + \varepsilon}$.

Across datasets, we observe consistent trends. On ETTh1, in 81.25% of feature-slice pairs (i.e., feature × temporal segment), $E[m \mid event] > E[m \mid non\text{-}event]$, and in 80.36% of the pairs, the retained ratio is also higher in event regions. Similarly, on ETTm2, these proportions are 75.03% and 63.39%, respectively. These results indicate that the gating mechanism is both responsive to structured changes and actively modulates the contribution of incremental signals, rather than uniformly re-weighting all increments. Notably, the gate tends to retain a larger fraction of increments in event regions while suppressing them in relatively stable regions, which are more likely to be dominated by high-frequency fluctuations introduced by differencing. This behavior is consistent with the intended effect of attenuating noise while preserving meaningful signal changes.

*Table 11.* Full ablation results. **Bold** indicates the best results.

| Datasets | Methods Metrics | Ours MSE | MAE | w/o L-Context MSE | MAE | w/o Gating MSE | MAE | RandContext MSE | MAE | w/o RelativePos MSE | MAE | GlobalPos MSE | MAE |
|---|---|---|---|---|---|---|---|---|---|---|---|---|---|
| ETTm1 | 96 | **0.310** | **0.354** | 0.319 | 0.360 | 0.318 | 0.361 | 0.314 | **0.354** | 0.316 | 0.358 | 0.316 | 0.355 |
|  | 192 | **0.354** | **0.379** | 0.355 | **0.379** | 0.360 | 0.383 | 0.356 | 0.381 | 0.355 | 0.381 | 0.356 | 0.381 |
|  | 336 | **0.381** | 0.402 | 0.386 | 0.402 | 0.389 | 0.406 | 0.385 | 0.402 | 0.383 | 0.404 | 0.384 | **0.401** |
|  | 720 | **0.437** | **0.437** | 0.444 | 0.439 | 0.443 | 0.439 | 0.447 | 0.440 | 0.444 | 0.441 | 0.443 | 0.438 |
|  | Avg | **0.371** | **0.393** | 0.376 | 0.395 | 0.378 | 0.398 | 0.375 | 0.394 | 0.375 | 0.396 | 0.375 | 0.394 |
| ETTm2 | 96 | **0.170** | **0.253** | 0.171 | 0.255 | **0.170** | **0.253** | 0.171 | 0.254 | 0.175 | 0.259 | 0.171 | 0.254 |
|  | 192 | 0.236 | 0.298 | 0.236 | 0.299 | 0.238 | 0.300 | **0.235** | 0.298 | 0.238 | 0.300 | 0.236 | **0.297** |
|  | 336 | **0.293** | **0.335** | 0.294 | 0.337 | 0.296 | 0.337 | 0.296 | 0.339 | 0.299 | 0.339 | 0.294 | 0.336 |
|  | 720 | 0.392 | **0.394** | 0.394 | 0.400 | 0.392 | **0.394** | **0.391** | 0.397 | 0.397 | 0.397 | **0.391** | 0.396 |
|  | Avg | **0.273** | **0.320** | 0.274 | 0.322 | 0.274 | 0.321 | **0.273** | 0.322 | 0.277 | 0.324 | **0.273** | 0.321 |
| ETTh1 | 96 | **0.368** | 0.393 | 0.372 | 0.399 | 0.374 | 0.398 | 0.374 | 0.394 | 0.370 | 0.393 | 0.371 | **0.392** |
|  | 192 | **0.417** | 0.426 | 0.428 | 0.428 | 0.424 | 0.426 | 0.420 | 0.427 | 0.419 | 0.427 | 0.421 | **0.424** |
|  | 336 | 0.455 | 0.446 | 0.456 | 0.447 | 0.458 | **0.443** | 0.461 | 0.451 | **0.451** | **0.443** | 0.472 | 0.450 |
|  | 720 | 0.467 | 0.468 | 0.482 | 0.473 | **0.466** | **0.461** | 0.480 | 0.474 | 0.485 | 0.470 | 0.471 | 0.462 |
|  | Avg | **0.427** | 0.433 | 0.434 | 0.437 | 0.430 | **0.432** | 0.434 | 0.437 | 0.431 | 0.433 | 0.434 | **0.432** |
| ETTh2 | 96 | **0.282** | **0.336** | 0.286 | 0.343 | 0.285 | 0.339 | 0.288 | 0.341 | 0.288 | 0.340 | 0.283 | 0.338 |
|  | 192 | **0.358** | **0.386** | 0.364 | 0.389 | 0.367 | 0.391 | 0.365 | 0.389 | 0.360 | 0.389 | 0.361 | 0.387 |
|  | 336 | **0.403** | **0.421** | 0.421 | 0.428 | 0.418 | 0.425 | 0.437 | 0.434 | 0.423 | 0.430 | 0.422 | 0.422 |
|  | 720 | 0.424 | 0.440 | 0.429 | 0.443 | **0.417** | **0.437** | 0.438 | 0.446 | 0.425 | 0.442 | 0.434 | 0.444 |
|  | Avg | **0.367** | **0.396** | 0.375 | 0.401 | 0.372 | 0.398 | 0.382 | 0.402 | 0.374 | 0.400 | 0.375 | 0.398 |
| Weather | 96 | **0.150** | **0.199** | 0.160 | 0.206 | 0.154 | 0.201 | 0.157 | 0.205 | 0.155 | 0.203 | 0.153 | 0.201 |
|  | 192 | **0.199** | **0.244** | 0.206 | 0.250 | 0.201 | 0.245 | 0.205 | 0.249 | 0.205 | 0.249 | 0.201 | 0.247 |
|  | 336 | **0.256** | **0.286** | 0.263 | 0.292 | 0.259 | 0.288 | 0.262 | 0.290 | 0.259 | 0.288 | 0.258 | 0.288 |
|  | 720 | **0.335** | **0.338** | 0.341 | 0.342 | 0.338 | 0.339 | 0.341 | 0.342 | 0.339 | 0.341 | 0.336 | 0.340 |
|  | Avg | **0.235** | **0.267** | 0.243 | 0.272 | 0.238 | 0.268 | 0.241 | 0.272 | 0.240 | 0.270 | 0.237 | 0.269 |
| Electricity | 96 | **0.141** | 0.240 | 0.150 | 0.241 | 0.148 | 0.245 | 0.151 | 0.242 | 0.149 | 0.247 | **0.141** | **0.239** |
|  | 192 | **0.156** | **0.253** | 0.163 | **0.253** | 0.166 | 0.263 | 0.164 | 0.254 | 0.163 | 0.261 | 0.162 | 0.262 |
|  | 336 | **0.172** | 0.272 | 0.180 | **0.271** | 0.180 | 0.277 | 0.181 | 0.272 | 0.185 | 0.280 | 0.179 | 0.277 |
|  | 720 | **0.206** | **0.298** | 0.216 | 0.304 | 0.209 | 0.300 | 0.216 | 0.304 | 0.213 | 0.302 | 0.215 | 0.311 |
|  | Avg | **0.169** | **0.266** | 0.177 | 0.267 | 0.176 | 0.271 | 0.178 | 0.268 | 0.178 | 0.273 | 0.174 | 0.272 |

*Table 12.* The ablation results of the standard positional encoding analysis. **Bold** indicates the best results.

| Datasets | Methods Metrics | Ours MSE | MAE | Positional-Encoding MSE | MAE | Datasets | Methods Metrics | Ours MSE | MAE | Positional-Encoding MSE | MAE |
|---|---|---|---|---|---|---|---|---|---|---|---|
| ETTh1 | 96 | **0.368** | **0.393** | 0.374 | 0.397 | ETTh2 | 96 | **0.282** | **0.336** | 0.283 | 0.337 |
|  | 192 | **0.417** | **0.426** | 0.426 | 0.428 |  | 192 | **0.358** | **0.386** | 0.364 | 0.388 |
|  | 336 | **0.455** | **0.446** | 0.467 | 0.467 |  | 336 | **0.403** | **0.421** | 0.413 | 0.425 |
|  | 720 | **0.467** | **0.468** | 0.508 | 0.482 |  | 720 | **0.424** | **0.440** | 0.427 | 0.444 |
|  | Avg | **0.427** | **0.433** | 0.444 | 0.443 |  | Avg | **0.367** | **0.396** | 0.371 | 0.398 |
| ETTm1 | 96 | **0.310** | **0.354** | 0.312 | 0.355 | ETTm2 | 96 | **0.170** | **0.253** | 0.172 | 0.256 |
|  | 192 | **0.354** | **0.379** | 0.356 | 0.382 |  | 192 | **0.236** | **0.298** | 0.236 | 0.299 |
|  | 336 | **0.381** | **0.402** | 0.383 | 0.404 |  | 336 | **0.293** | **0.335** | 0.294 | 0.337 |
|  | 720 | **0.437** | **0.437** | 0.440 | 0.438 |  | 720 | **0.392** | **0.394** | 0.395 | 0.398 |
|  | Avg | **0.371** | **0.393** | 0.373 | 0.395 |  | Avg | **0.273** | **0.320** | 0.275 | 0.323 |

*Table 13.* Full portability results. **Bold** indicates the best results.

| Datasets | Methods | DLinear | | | | iTransformer | | | | PatchTST | | | |
|---|---|---|---|---|---|---|---|---|---|---|---|---|---|
| | | Original | | +L-Context | | Original | | +L-Context | | Original | | +L-Context | |
| | Metrics | MSE | MAE | MSE | MAE | MSE | MAE | MSE | MAE | MSE | MAE | MSE | MAE |
| ETTh1 | 96 | **0.386** | 0.400 | 0.388 | **0.399** | 0.394 | 0.409 | **0.390** | **0.406** | 0.414 | 0.419 | **0.379** | **0.395** |
| | 192 | **0.437** | 0.432 | 0.443 | **0.430** | 0.448 | 0.441 | **0.438** | **0.437** | 0.460 | 0.445 | **0.423** | **0.425** |
| | 336 | **0.481** | 0.459 | 0.484 | **0.448** | 0.491 | 0.464 | **0.489** | **0.461** | 0.501 | 0.466 | **0.466** | **0.446** |
| | 720 | 0.519 | 0.516 | **0.490** | **0.474** | **0.519** | 0.502 | 0.522 | **0.501** | **0.500** | **0.488** | 0.518 | 0.502 |
| | Avg | 0.456 | 0.452 | **0.451** | **0.438** | 0.463 | 0.454 | **0.460** | **0.451** | 0.469 | 0.455 | **0.447** | **0.442** |
| ETTm1 | 96 | 0.345 | 0.372 | **0.340** | **0.365** | 0.341 | 0.376 | **0.334** | **0.370** | 0.329 | 0.367 | **0.322** | **0.358** |
| | 192 | 0.380 | 0.389 | **0.379** | **0.384** | 0.381 | 0.395 | **0.373** | **0.390** | **0.367** | **0.385** | 0.368 | 0.388 |
| | 336 | 0.413 | 0.413 | **0.412** | **0.407** | 0.417 | 0.418 | **0.409** | **0.413** | 0.399 | 0.410 | **0.396** | **0.405** |
| | 720 | **0.474** | 0.453 | **0.474** | **0.439** | 0.487 | 0.456 | **0.48** | **0.453** | **0.454** | **0.439** | 0.458 | 0.442 |
| | Avg | 0.403 | 0.407 | **0.401** | **0.399** | 0.407 | 0.411 | **0.399** | **0.407** | 0.387 | 0.400 | **0.386** | **0.398** |
| ECL | 96 | 0.197 | 0.282 | **0.175** | **0.273** | **0.148** | **0.240** | 0.154 | 0.251 | 0.181 | 0.270 | **0.158** | **0.261** |
| | 192 | 0.196 | 0.285 | **0.187** | **0.284** | **0.162** | **0.253** | 0.166 | 0.262 | 0.188 | **0.274** | **0.171** | 0.274 |
| | 336 | 0.209 | 0.301 | **0.200** | **0.299** | 0.178 | **0.269** | **0.175** | 0.272 | 0.204 | 0.293 | **0.184** | **0.287** |
| | 720 | 0.245 | 0.333 | **0.230** | **0.323** | 0.225 | 0.317 | **0.210** | **0.302** | 0.246 | 0.324 | **0.224** | **0.320** |
| | Avg | 0.212 | 0.300 | **0.198** | **0.295** | 0.178 | **0.270** | **0.176** | 0.272 | 0.205 | 0.290 | **0.184** | **0.286** |
| Weather | 96 | 0.196 | 0.255 | **0.184** | **0.228** | 0.174 | 0.214 | **0.165** | **0.209** | **0.177** | 0.218 | 0.179 | **0.216** |
| | 192 | 0.237 | 0.296 | **0.233** | **0.268** | 0.221 | 0.254 | **0.215** | **0.253** | 0.225 | 0.259 | **0.215** | **0.256** |
| | 336 | **0.283** | 0.335 | 0.285 | **0.304** | 0.278 | 0.296 | **0.272** | **0.295** | 0.278 | **0.297** | **0.276** | 0.297 |
| | 720 | **0.345** | 0.381 | 0.358 | **0.352** | 0.358 | 0.349 | **0.354** | **0.348** | **0.354** | **0.348** | 0.355 | 0.350 |
| | Avg | **0.265** | 0.317 | **0.265** | **0.288** | 0.258 | 0.278 | **0.251** | **0.276** | 0.259 | 0.281 | **0.256** | **0.280** |

*Table 14.* The results of the lag error analysis. **Bold** indicates the best results.

| Datasets | Methods | L-Drive | | TimeBridge | | iTransformer | |
|---|---|---|---|---|---|---|---|
| | Metrics | TailAUC | ExcessAUC | TailAUC | ExcessAUC | TailAUC | ExcessAUC |
| ETTh1 | 96 | **21.53** | **9.94** | 25.42 | 11.03 | 25.05 | 11.34 |
| | 192 | **44.19** | **11.36** | 55.20 | 15.09 | 51.19 | 13.74 |
| | 336 | **81.58** | **17.77** | 105.48 | 24.24 | 101.47 | 21.87 |
| | 720 | **236.44** | **110.81** | 293.80 | 132.42 | 265.30 | 128.34 |
| | Avg | **95.94** | **37.47** | 119.98 | 45.69 | 110.75 | 43.82 |
| EPF | NP | **2.81** | **1.30** | 6.20 | 2.83 | 16.11 | 6.97 |
| | PJM | **1.70** | **0.76** | 6.71 | 2.71 | 8.52 | 3.30 |
| | BE | **2.70** | **1.34** | 2.83 | 1.48 | 8.36 | 4.00 |
| | FR | 2.47 | 1.17 | **2.33** | **1.12** | 7.39 | 3.21 |
| | DE | **2.19** | **0.91** | 2.36 | 1.05 | 8.20 | 3.20 |
| | Avg | **2.37** | **1.10** | 4.08 | 1.84 | 9.71 | 4.14 |
| Weather | 96 | **5.88** | **3.53** | 6.25 | 3.82 | 7.66 | 4.47 |
| | 192 | **11.65** | **6.85** | 12.76 | 7.37 | 14.83 | 8.13 |
| | 336 | **22.47** | **12.40** | 24.06 | 14.54 | 27.95 | 16.20 |
| | 720 | **60.80** | **30.57** | 61.93 | 32.20 | 71.53 | 40.35 |
| | Avg | **25.20** | **13.34** | 26.25 | 14.48 | 30.49 | 17.29 |
| Solar | 96 | 6.43 | 4.22 | **5.98** | 4.50 | 7.24 | **3.61** |
| | 192 | **11.43** | 6.87 | 11.58 | 8.15 | 14.91 | **6.48** |
| | 336 | **21.32** | **14.60** | 22.30 | 15.32 | 28.01 | 15.59 |
| | 720 | **51.60** | **33.77** | 53.32 | 35.86 | 65.85 | 36.95 |
| | Avg | **22.69** | **14.86** | 23.29 | 15.96 | 29.00 | 15.66 |

*Table 15.* Full hyperparameter sensitivity analysis for $d_{\text{pos}}$.

| Datasets | $d_{\text{pos}}$ | 2 | | 4 | | 6 | | 8 | | Datasets | $d_{\text{pos}}$ | 2 | | 4 | | 6 | | 8 | |
|---|---|---|---|---|---|---|---|---|---|---|---|---|---|---|---|---|---|---|---|
| | Metrics | MSE | MAE | MSE | MAE | MSE | MAE | MSE | MAE | | Metrics | MSE | MAE | MSE | MAE | MSE | MAE | MSE | MAE |
| ETTm1 | 96 | 0.310 | 0.354 | 0.311 | 0.355 | 0.315 | 0.357 | 0.319 | 0.358 | ETTm2 | 96 | 0.170 | 0.253 | 0.172 | 0.255 | 0.172 | 0.255 | 0.171 | 0.254 |
| | 192 | 0.354 | 0.379 | 0.364 | 0.387 | 0.354 | 0.381 | 0.354 | 0.381 | | 192 | 0.236 | 0.298 | 0.237 | 0.298 | 0.237 | 0.300 | 0.236 | 0.298 |
| | 336 | 0.381 | 0.402 | 0.385 | 0.403 | 0.385 | 0.404 | 0.384 | 0.403 | | 336 | 0.293 | 0.335 | 0.293 | 0.336 | 0.297 | 0.338 | 0.296 | 0.339 |
| | 720 | 0.437 | 0.437 | 0.449 | 0.442 | 0.439 | 0.436 | 0.443 | 0.439 | | 720 | 0.392 | 0.394 | 0.393 | 0.398 | 0.396 | 0.399 | 0.394 | 0.398 |
| | Avg | 0.371 | 0.393 | 0.377 | 0.397 | 0.373 | 0.394 | 0.375 | 0.396 | | Avg | 0.273 | 0.320 | 0.274 | 0.322 | 0.276 | 0.323 | 0.274 | 0.322 |
| ETTh1 | 96 | 0.368 | 0.393 | 0.367 | 0.393 | 0.367 | 0.392 | 0.371 | 0.394 | ETTh2 | 96 | 0.282 | 0.336 | 0.287 | 0.340 | 0.287 | 0.339 | 0.282 | 0.339 |
| | 192 | 0.417 | 0.426 | 0.419 | 0.426 | 0.418 | 0.426 | 0.417 | 0.427 | | 192 | 0.358 | 0.386 | 0.363 | 0.388 | 0.363 | 0.388 | 0.360 | 0.387 |
| | 336 | 0.455 | 0.446 | 0.471 | 0.446 | 0.463 | 0.446 | 0.459 | 0.450 | | 336 | 0.403 | 0.421 | 0.416 | 0.425 | 0.416 | 0.426 | 0.424 | 0.427 |
| | 720 | 0.467 | 0.468 | 0.455 | 0.457 | 0.476 | 0.472 | 0.467 | 0.461 | | 720 | 0.424 | 0.440 | 0.423 | 0.439 | 0.421 | 0.439 | 0.423 | 0.439 |
| | Avg | 0.427 | 0.433 | 0.428 | 0.430 | 0.431 | 0.434 | 0.429 | 0.433 | | Avg | 0.367 | 0.396 | 0.372 | 0.398 | 0.372 | 0.398 | 0.372 | 0.398 |
| Weather | 96 | 0.150 | 0.199 | 0.153 | 0.201 | 0.154 | 0.202 | 0.152 | 0.199 | Electricity | 96 | 0.141 | 0.240 | 0.143 | 0.242 | 0.143 | 0.242 | 0.144 | 0.242 |
| | 192 | 0.199 | 0.244 | 0.200 | 0.245 | 0.201 | 0.245 | 0.201 | 0.246 | | 192 | 0.156 | 0.253 | 0.158 | 0.257 | 0.159 | 0.257 | 0.161 | 0.260 |
| | 336 | 0.256 | 0.286 | 0.261 | 0.289 | 0.258 | 0.286 | 0.258 | 0.288 | | 336 | 0.172 | 0.272 | 0.178 | 0.278 | 0.175 | 0.276 | 0.180 | 0.280 |
| | 720 | 0.335 | 0.338 | 0.342 | 0.343 | 0.338 | 0.340 | 0.338 | 0.340 | | 720 | 0.206 | 0.298 | 0.210 | 0.301 | 0.202 | 0.296 | 0.219 | 0.304 |
| | Avg | 0.235 | 0.267 | 0.239 | 0.270 | 0.238 | 0.268 | 0.237 | 0.269 | | Avg | 0.169 | 0.266 | 0.172 | 0.269 | 0.170 | 0.268 | 0.176 | 0.271 |

*Table 16.* Full hyperparameter sensitivity analysis for Conv1d kernel size.

| Datasets | KernelSize | 2 | | 3 | | 4 | | 6 | | Datasets | KernelSize | 2 | | 3 | | 4 | | 6 | |
|---|---|---|---|---|---|---|---|---|---|---|---|---|---|---|---|---|---|---|---|
| | Metrics | MSE | MAE | MSE | MAE | MSE | MAE | MSE | MAE | | Metrics | MSE | MAE | MSE | MAE | MSE | MAE | MSE | MAE |
| ETTh1 | 96 | 0.374 | 0.394 | 0.368 | 0.393 | 0.371 | 0.395 | 0.375 | 0.396 | ETTh2 | 96 | 0.284 | 0.340 | 0.282 | 0.336 | 0.286 | 0.339 | 0.281 | 0.336 |
| | 192 | 0.423 | 0.425 | 0.417 | 0.426 | 0.418 | 0.428 | 0.421 | 0.427 | | 192 | 0.368 | 0.391 | 0.358 | 0.386 | 0.364 | 0.388 | 0.366 | 0.392 |
| | 336 | 0.456 | 0.445 | 0.455 | 0.446 | 0.464 | 0.445 | 0.481 | 0.454 | | 336 | 0.422 | 0.427 | 0.403 | 0.421 | 0.413 | 0.423 | 0.438 | 0.443 |
| | 720 | 0.470 | 0.461 | 0.455 | 0.457 | 0.479 | 0.475 | 0.489 | 0.475 | | 720 | 0.455 | 0.457 | 0.421 | 0.439 | 0.429 | 0.443 | 0.428 | 0.443 |
| | Avg | 0.431 | 0.431 | 0.424 | 0.430 | 0.433 | 0.436 | 0.441 | 0.438 | | Avg | 0.382 | 0.404 | 0.366 | 0.396 | 0.373 | 0.398 | 0.378 | 0.404 |
| ETTm1 | 96 | 0.318 | 0.359 | 0.310 | 0.354 | 0.314 | 0.356 | 0.321 | 0.360 | ETTm2 | 96 | 0.172 | 0.255 | 0.170 | 0.253 | 0.172 | 0.255 | 0.171 | 0.255 |
| | 192 | 0.357 | 0.385 | 0.354 | 0.379 | 0.355 | 0.383 | 0.355 | 0.383 | | 192 | 0.236 | 0.299 | 0.236 | 0.298 | 0.236 | 0.298 | 0.236 | 0.298 |
| | 336 | 0.382 | 0.402 | 0.381 | 0.402 | 0.383 | 0.404 | 0.382 | 0.403 | | 336 | 0.296 | 0.337 | 0.293 | 0.335 | 0.298 | 0.341 | 0.294 | 0.338 |
| | 720 | 0.442 | 0.438 | 0.437 | 0.437 | 0.445 | 0.442 | 0.441 | 0.440 | | 720 | 0.394 | 0.397 | 0.392 | 0.394 | 0.389 | 0.393 | 0.393 | 0.397 |
| | Avg | 0.375 | 0.396 | 0.371 | 0.393 | 0.374 | 0.396 | 0.375 | 0.397 | | Avg | 0.274 | 0.322 | 0.273 | 0.320 | 0.274 | 0.322 | 0.274 | 0.322 |

*Table 17.* Training and inference time on Weather.

| Metrics | Ours | CrossLinear | FilterTS | TimeXer | PatchTST | TimesNet |
|---|---|---|---|---|---|---|
| Training Time | 23.887 | 33.25 | 36.552 | 29.584 | 381.879 | 335.324 |
| Inference Time | 7.645 | 12.144 | 5.244 | 3.924 | 81.902 | 99.497 |

