# OpenReview forum: "L-Drive: Beyond a Single Mapping—Latent Context Drives Time Series Forecasting"
_ICML.cc/2026/Conference — ICML 2026 regular_

### Official Review · Reviewer_nF9B · 2026-03-10

**Soundness:** 3
**Presentation:** 1
**Significance:** 2
**Originality:** 2
**Overall Recommendation:** 4
**Confidence:** 4

**Summary:**

This paper identifies a systematic response lag problem in Direct-Mapping multivariate time series forecasting models: predictions fail to adapt promptly to distribution shifts or regime changes, instead trailing behind the underlying dynamics. The authors argue that this lag stems from a recursive inertial structure inherent to deployed models, and draw an analogy to the adaptive expectations model from economics to formalize this phenomenon.

To address this, L-Drive is proposed, consisting of two components:

- **L-Context Generator**: Normalizes the input, applies a Conv1D with residual addition, computes first-order differences, then concatenates the original and differenced signals and passes this through a gating MLP (sigmoid output) to produce a selective gate. Importantly, the gate is applied only to the differenced signal rather than to the full concatenated input, and this gated difference signal is fed into a GRU (internal sigmoid/tanh activations) to produce the time-evolving latent context.

- **Struct-Aided Predictor**: Fuses L-Context with the original sequence, splits the result into non-overlapping patches, and applies patch-shared relative positional basis functions. This is intended to strengthen intra-segment structural pattern modeling while reducing overfitting caused by memorizing absolute positions.

The key claims are:

1. Direct-Mapping models exhibit systematic response lag at regime switches, formalized via an adaptive expectations analogy with an explicit residual lag upper bound derived from the recursive inertial structure.
2. L-Context can be used in a model-agnostic manner. The paper demonstrates that injecting it into existing architectures such as DLinear, iTransformer, and PatchTST yields performance improvements without architectural modification.
3. Patch-shared relative positional encoding reduces overfitting from absolute position memorization and stabilizes training.

Experiments use 8 datasets (ETTm1/m2, ETTh1/h2, ECL, Weather, Solar, EPF) with lookback window 96 and prediction horizons {96, 192, 336, 720}, compared against 9 baselines spanning MLP, Transformer, CNN, and time-frequency paradigms.

**Compliance With Llm Reviewing Policy:**

Affirmed.

**Final Justification:**

Thank you for the detailed responses and supplementary analyses provided both directly and through the cross-referenced replies to Reviewer 4rgi.

**W1 (Theory-experiment mismatch)**

The three-form fitting experiment is a substantive addition. Demonstrating that the full form (Eq.2 proxy) achieves the highest $R^2$ while the inertia-only form confirms strong dependence on past predictions, and the update-only form proves unstable, provides meaningful empirical support that Eq.2 captures the local rollout behavior of single-forward-pass baselines. I accept this as a faithful local approximation rather than a purely conceptual analogy.

**W2, W4, W6, W11**

The three conditions justifying Eq.21, the multi-run statistics for ablations, the extended lag error results across electricity, weather, and energy domains, and the event-conditioned gating analysis all directly address my original concerns. These are genuine improvements to the empirical grounding of the paper.

**W3 (L-Context as increment estimator)**

A gap remains between "L-Context can be approximated as a function of the difference sequence" and the direct estimation claim in Eq.(12). The Lipschitz-based proof and dCor evidence together support the structural interpretation, but the direct estimation framing in Eq.(12) remains stronger than what the evidence strictly demonstrates. I would encourage the authors to adjust the language in the final manuscript to more precisely reflect what is proven versus what is assumed.

**W5 (Struct-Aided Predictor motivation)**

The explanation connecting relative positional bases to location-agnostic change pattern learning is more convincing than the original framing. However, the motivation remains partially generic relative to the paper's core change-aware narrative. This is a presentation concern rather than a fundamental flaw.

**Overall**

The authors have responded substantively to the most critical concerns, particularly W1, which was the most fundamental issue I raised. The remaining gaps in W3 and W5 are real but do not undermine the paper's core contribution. Given the overall quality of the response and the genuine improvements provided, I am raising my Overall Recommendation to 4 and maintaining my Confidence at 4.

**Key Questions For Authors:**

- **Q1.** How does Eq. (2) apply to the DMSF baselines in the experiments, which produce the full forecast horizon in a single forward pass with no autoregressive feedback? If the lag in these models actually comes from post-switch samples gradually entering the fixed lookback window rather than recursive inertia, how does L-Context address that specific mechanism?

- **Q2.** Several formula-related issues seem to require clarification. In the transition from Eq. (12) to Eq. (13), L-Context is treated as estimating $\Delta g_t$, but L-Context is simply the GRU hidden state at each step, and there is no explicit training signal pushing it toward increment estimation, so this correspondence seems to be assumed rather than demonstrated. Could the authors provide empirical evidence, such as a correlation analysis between L-Context outputs and ground-truth increments, to support this interpretation? Additionally, in the transition from Eq. (20) to Eq. (22) in the appendix, the model state is approximated as the prediction value itself. Could the authors provide a more rigorous theoretical justification for why this approximation holds, and under what conditions it may break down?

- **Q3.** Were the ablation results in Table 2 averaged over multiple runs? Given the magnitude of differences (0.001 level), single-run comparisons seem unreliable. How do the authors explain cases where Struct-Aided Predictor ablations produce larger drops than the L-Context ablation?

- **Q4.** Why is CrossLinear excluded from the lag analysis in Table 4? How sensitive are TailAUC/ExcessAUC to the choice of percentile threshold and neighborhood window size, and could these parameters be tuned to favor the proposed method?

- **Q5.** Is there a principled reason why the Struct-Aided Predictor's patch-based design is specifically suited to change-aware forecasting, or is it a general-purpose forecasting improvement? If it reduces overfitting by avoiding absolute position memorization, is there empirical evidence that absolute position memorization is actually a problem for the baseline architectures compared? How does the proposed approach compare to standard positional encoding ablations in existing work?

- **Q6.** The paper draws a core theoretical connection between deployed deep learning forecasting models and the adaptive expectations model from economics. The adaptive expectations model was derived under specific assumptions (monotone expectations, scalar quantities, linear update rules), whereas deep learning forecasting models operate in high-dimensional spaces with complex non-linear dynamics. Is there a formal theoretical basis or a known connection in the literature supporting this analogy? Without such a foundation, the theoretical framework risks being more metaphorical than analytical.

**Limitations:**

The paper does not have a dedicated limitations section. Several important limitations go unacknowledged. First and most critically, there is a fundamental mismatch between the theoretical motivation and the experimental setup (W1): the formal analysis in Section 3 is built on a recursive rollout assumption that does not hold for the DMSF models used as baselines, all of which produce the full forecast horizon in a single forward pass. This means the lag upper bound derived in Section 4.3, while internally consistent, is grounded in a deployment model that does not describe what the evaluated architectures actually do. Second, the approximation $s_{t-1} \approx \hat{y}_{t-1}$ in Appendix A collapses a high-dimensional hidden state to a scalar prediction, and no justification is provided for when this simplification is valid or what theoretical guarantees are lost by making it (W2). Third, the claim that L-Context functions as an estimator of the increment $\Delta g_t$ is not enforced by any component of the training objective; the GRU is trained solely under MSE loss, so the theoretical narrative connecting L-Context to increment estimation relies on an unverified assumption rather than a design guarantee (W3). Fourth, the gating mechanism's claimed ability to suppress high-frequency noise introduced by differencing is not supported by theoretical analysis or targeted empirical verification; it is unclear why the proposed concatenation-based gating would selectively attenuate noise rather than uniformly re-weight all incremental components (W11). Fifth, the Struct-Aided Predictor is introduced without a clearly defined problem motivation specific to change-aware forecasting; unlike the L-Context Generator, which addresses an explicitly stated failure mode, the Struct-Aided Predictor is motivated only by generic overfitting concerns, making it difficult to determine whether it is an essential contribution or a general-purpose performance add-on (W5).

**Strengths And Weaknesses:**

**Strengths**

- **S1.** In my view, the problem formulation is one of the strongest aspects of this paper. Explicitly identifying response lag as a distinct failure mode of Direct-Mapping models, separate from general distribution shift, and giving it a concrete formal characterization is a meaningful contribution to the community's understanding of when and why current forecasting models fail. The synthetic Mode 1/2 dataset in Figure 1 makes this failure mode immediately visible and easy to reason about. This kind of clear problem definition appears to be a genuinely valuable contribution regardless of the technical details of the proposed solution.


- **S2.** The mathematics in Section 4.3 and Appendix B appear internally consistent. The derivation from Eq. (10) to Eq. (17) seems to hold up, and the resulting upper bound gives a tidy formal statement of L-Context's intended role. My concerns about the assumptions underlying this theory are noted separately in W1-W3, but given those assumptions, the derivation itself looks correct.

- **S3.** The portability results in Table 3 appear meaningful from the perspective of time series learning methodology. Obtaining consistent gains across DLinear, iTransformer, and PatchTST, three architectures with very different inductive biases, without modifying their internals suggests that L-Context may capture information complementary to what these architectures already learn, rather than simply compensating for a weakness specific to one model.

---

**Weaknesses**

- **W1.** The entire theoretical motivation in Section 3 is built around Eq. (2), which models predictions as a recursive rollout where the previous prediction feeds into the next step. The problem is that essentially every baseline in this paper, including PatchTST, iTransformer, DLinear, CrossLinear, FilterTS, PatchMLP, TimesNet, TimeXer, and TimeBridge, takes a fixed window and outputs the full horizon in a single forward pass. There is no autoregressive loop, so Eq. (2) does not describe what these models do during inference. The lag these models actually exhibit appears to be more naturally explained by the fact that after a regime switch, new post-switch observations only gradually fill up the fixed lookback window, so the model's effective input distribution shifts slowly. That is a structurally different mechanism from what the paper formalizes. The core idea of using a latent context to capture change dynamics is, in my view, genuinely interesting. However, the formal justification appears to be motivated by a problem that does not straightforwardly arise in the model architectures actually used in the experiments. This gap between the theoretical motivation and the experimental setup seems to be the most fundamental issue in the paper.

- **W2.** Appendix A approximates the model's internal state as $s_{t-1} \approx \hat{y}_{t-1}$, collapsing a high-dimensional hidden representation to a scalar output. For any model with meaningful capacity, the hidden state carries substantially more information than the scalar output. I think this approximation would need a more careful justification, specifically regarding under what conditions it is reasonable and what theoretical cost it incurs.

- **W3.** In Eq. (12), L-Context is cast as estimating $\Delta g_t$, but the GRU is trained purely under MSE loss with no component of the objective explicitly pushing L-Context toward increment estimation. The theoretical narrative in Eqs. (12)-(13), while the algebra appears correct (see S2), therefore reads more as a post-hoc rationalization than a design guarantee. An empirical check, such as how well L-Context outputs correlate with $\Delta g_t$ on held-out data, would help substantiate this claim.

- **W4.** The ablation gaps in Table 2 are very small. On ETTm2, the difference between the full model and the w/o L-Context variant is reported as 0.001 MSE. Without multiple runs and error bars, it seems difficult to conclude that these differences are real rather than noise. No variance estimates or significance tests appear anywhere in the ablation section.

- **W5.** Looking more carefully at Table 2, the gains from removing the RelativePos and GlobalPos components sometimes appear larger than the gains from removing L-Context. If the Struct-Aided Predictor components do more work than L-Context on some datasets, this seems awkward given the paper's framing. More fundamentally, the Struct-Aided Predictor does not appear to have a clear problem statement of its own. The L-Context Generator is motivated by an explicitly defined failure mode (response lag), but the Struct-Aided Predictor is introduced primarily with the justification that it "reduces overfitting caused by memorizing specific positions." This is a generic motivation not specific to the change-aware forecasting setting the paper targets, and the module ends up feeling like a general-purpose forecasting head rather than a component motivated by the core problem.

- **W6.** The Lag Error Analysis in Section 5.4.3 appears to have a circularity problem. Change events are detected using first-order differences, which is the same operation that drives the L-Context Generator, so the evaluation methodology seems structurally aligned with the method being evaluated. Results are also only shown for a single dataset (ETTh1), CrossLinear is omitted from Table 4 without explanation, and the neighborhood window size and minimum gap parameter are not specified in the main text.

- **W7.** The Related Work focuses on channel-mixing vs. independence and time-frequency representations, which is largely orthogonal to the paper's actual argument. More importantly, I think it is worth clarifying the relationship between response lag and distribution shift more carefully. Distribution shift refers to a statistical change in the data-generating process and is broadly addressed through domain adaptation, test-time adaptation, or continual learning methods. Response lag, as defined in the paper, appears to be a narrower phenomenon: the model's temporal delay in responding to a sudden regime switch. In my view, these are related but conceptually distinct, in that reducing response lag does not obviously address distribution shift in general, and distribution shift does not always manifest as response lag. Given this, I think it would be more appropriate to situate L-Drive in relation to the literature on training strategies for time-series models, rather than framing it primarily in contrast to structural design choices such as channel mixing or time–frequency decompositions.

- **W8.** In Section 4.1, the text states that the Conv1D output is added back to the input "in a residual manner," but no formula is provided for this step. More broadly, the gating MLP $g(\cdot)$ is described only as an "MLP-based gating network" without specifying its number of layers, hidden dimensions, intermediate activation functions, or the output projection required to reduce the input dimensionality for the element-wise product to be valid. A more detailed description of the gating MLP structure is necessary for reproducibility.

- **W9.** Two notation issues are worth noting. First, Eq. (9) includes $+\varepsilon$ as an additive term in the denormalization formula, which appears inconsistent with Eq. (4) where $\varepsilon$ is defined as a small constant in the denominator for numerical stability. Second, in Eq. (6), the gating MLP $g(\cdot)$ takes the concatenated input as input, but the output dimension is never stated. Since $h_t = m_t \odot \Delta x'_t$ requires the gate to match the dimension of the differenced signal, the implicit dimension reduction inside the gating MLP should be made explicit. More broadly, the notation throughout the paper would benefit from a systematic review and refinement for consistency.

- **W10.** The hyperparameter sensitivity analysis only evaluates $d_{pos}$. This is probably not the most sensitive parameter in the model, as the GRU hidden dimension, patch size $P$, and Conv1D kernel size are all likely more consequential, and none of them are analyzed.

- **W11.** The paper claims that the gating mechanism suppresses "inconsistent or weakly related incremental components, thereby reducing the high-frequency noise that differencing may introduce." However, the theoretical or empirical basis for this claim appears insufficient. It is not clear why concatenating the original and differenced signals into a gating MLP would specifically suppress noise rather than simply re-weighting all incremental components uniformly. I think a more rigorous analysis or targeted ablation would be needed to support this claim.

- **W12.** Several presentation issues are worth noting. First, there appear to be spacing errors in the text (e.g., "here,ε" should be "here, ε"), and the writing contains expressions that feel unpolished in places. Second, figure text is in some places too small to read clearly, and at least one figure contains a clear typographic error ("Ground True" should be "Ground Truth"). Third, several concepts and terms used in the paper (e.g., adaptive expectations, the proposed evaluation metrics) lack citations to their original sources; if these are not novel to this paper, references should be provided.

---

> ### Author Rebuttal · Authors · 2026-03-31
>
> We thank the reviewer for the insightful feedback.
>
> W1
>
> We clarify that Eq.2 is not meant to literally describe the inference graph of each architecture, but to provide a local reduced-form description of the prediction inertia commonly observed around regime switches. Specifically, rather than reproducing the exact forward computation of different Direct-Mapping architectures, Eq.2 uses a low-order local approximation of adjacent predictions near the switch to isolate the core inertia pattern. In other words, Eq.2 characterizes the local reduced-form dynamics of adjacent predictions, not the model’s literal rollout process.
>
> The reviewer also suggests that “the lag is more naturally explained by the fact that after a regime switch, new post-switch observations only gradually fill up the fixed lookback window, so the model's effective input distribution shifts slowly”. We find this perspective important and insightful, and fully consistent with our main point: whether the lag arises from explicit recursive rollout or from the gradual removal of pre-switch information in a fixed lookback window, the common feature is that information from the old regime does not vanish instantly, but continues to affect predictions for several steps.
>
> Thus, the fixed-lookback-window explanation does not conflict with our main motivation; rather, it provides a more concrete account, from the perspective of one-shot Direct-Mapping, of why models relying solely on a unified mapping in observation space tend to retain the previous level, slope, or local correlation structure after a regime switch. This also supports our motivation for introducing latent Context: instead of relying only on gradual window replacement, we want the model to explicitly represent whether the underlying dynamics have changed, so that it can align with the new regime more rapidly. We will revise it to enhance readability.
>
> W2
>
> We clarify that the argument in Appendix A does not rely on simply replacing the previous full state with the previous prediction. We still start from the general full-state representation (Eq.19,20). On this basis, to obtain a one-dimensional reduced-form recursion in the prediction space for analyzing prediction inertia, we define Eq.21. The key step is not to equate the full state with the prediction, but to average over the part of the state information that remains unobserved even after conditioning on the current input and the previous prediction. In this way, we obtain a conditional-mean reduced form that depends only on the current input and the previous prediction.
>
> Put differently, we do not deny that the high-dimensional state contains richer information than the output prediction itself. On the contrary, it is precisely because the previous full state in complex sequence models is generally not uniquely pinned down by the previous prediction that we introduce a conditional expectation operator. This allows us to project the full-state representation onto a prediction-space reduced form. The purpose of this formulation is exactly to avoid mischaracterizing the prediction-space recursion as hidden-state identification. More precisely, this reduced form is not meant to recover the full hidden-state dynamics, but to provide an interpretable and linearizable local approximation of output inertia around regime switches. It is therefore tailored to the local analysis in our paper, and the subsequent Taylor expansion and contraction condition are both stated for this local operating region.
>
> W3
>
> Please refer to our Reply on Theory to Reviewer 4rgi.
>
> W4
>
> Please refer to the second point in our Reply on Experiment to Reviewer 4rgi.
>
> W5
>
> From the full results in Table 7, aggregating across all datasets, the performance drop caused by removing L-Context remains larger than that caused by removing RelativePos or GlobalPos. This indicates that L-Context plays a consistently stable role as the core change-aware representation. By contrast, the benefit of the Struct-Aided Predictor is more reflected in enabling change information to be translated more stably into responses to local dynamics. We therefore view the two as complementary rather than competing: the former addresses change perception and representation, while the latter determines how such information is structurally converted into outputs. In addition, we include standard positional-encoding ablations, and our method still performs better overall. The maximum improvement across all results is 8.09%.
>
> https://anonymous.4open.science/r/LRe-CE4F/pos.md
>
> W6
>
> Please refer to our Reply to Question 2 to Reviewer SEF1.
>
> W7,8,9,11,12
>
> All typos, phrasing issues will be corrected and the discussion on related work will be expanded in the revision.
>
> W10
>
> We added the parameter sensitivity analysis regarding the Conv1D.
>
> https://anonymous.4open.science/r/LRe-CE4F/kernel.md
>
> We hope these clarifications and additional results help address the reviewer’s concerns.

---

> > ### Author Rebuttal · Reviewer_nF9B · 2026-04-01
> >
> > I thank the authors for their thoughtful rebuttal, both the direct response and the supplementary materials provided to other reviewers (dCor analysis, multi-run standard deviations, lag error parameter details). These demonstrate genuine effort and engagement with the concerns raised.
> >
> > Before detailing my assessment, I note that my evaluation prioritizes two aspects: (1) whether the problem formulation and methodology are well-aligned, with the theoretical framework demonstrably applicable to the models actually tested, and (2) whether the problem formulation itself is convincingly motivated and properly scoped. My remaining concerns span both dimensions.
> >
> > **W1/Q1 (Theory–experiment mismatch):** The reframing of Eq.(2) as a "local reduced-form description of prediction inertia" rather than the literal inference graph is a reasonable conceptual move. The acknowledgment that the fixed-lookback-window explanation is consistent with the paper's motivation is constructive. However, this reframing transforms the theoretical contribution from a formal derivation describing the tested models to an analogy capturing a shared qualitative property. Sections 3 and 4.3 build heavily on the recursive form to motivate L-Context and derive the lag upper bound (Eq.17), so their strength depends on how well Eq.(2) approximates the actual behavior of single-forward-pass baselines — which remains empirically undemonstrated.
> >
> > **W2/Q2 (State-to-scalar approximation):** The conditional-expectation projection (Eq.21) is mathematically well-defined and addresses the concern. The remaining gap is quantitative: under what conditions is this projection tight enough for the Taylor expansion and contraction argument to hold meaningfully?
> >
> > **W3 (L-Context as increment estimator):** The distance correlation analysis is a valuable addition. The consistently high dCor values (0.782–0.892) across all seven datasets with small standard deviations provide meaningful empirical evidence that L-Context captures variation-related structural information. This substantially mitigates the concern that the increment interpretation is purely post-hoc. However, dCor measures structural similarity between distance matrices, which supports "L-Context encodes change-related structure" but is weaker than demonstrating "L-Context directly estimates $\Delta g_t$" as claimed in Eq.(12). The gap between structural correlation and direct estimation remains.
> >
> > **W4 (Ablation significance):** The reporting of multi-run means and standard deviations is appreciated and directly addresses the core concern. While I cannot verify the linked results in full, the authors' claim of low and stable standard deviations, if accurate, would substantially strengthen the ablation evidence.
> >
> > **W5 (Struct-Aided Predictor):** The complementary framing and the positional encoding comparison are noted. However, the Struct-Aided Predictor's motivation remains generic ("reduce overfitting from position memorization") rather than specific to the change-aware setting, which weakens its integration into the paper's core narrative.
> >
> > **W6 (Lag error analysis):** The clarification that change events are detected from ground-truth (not model outputs) using a model-agnostic definition, with parameters (window=25, gap=24) kept consistent across all baselines, substantially addresses the circularity concern. However, results remain limited to ETTh1. Additional datasets would strengthen the analysis.
> >
> > **W11 (Gating mechanism):** No evidence was provided for the selective noise suppression claim, either in the direct response or in the cross-referenced replies.
> >
> > **Overall:** The rebuttal, together with the cross-referenced materials, represents a more substantial response than initially apparent. The dCor analysis (W3), multi-run statistics (W4), and lag error methodology clarification (W6) are genuine improvements. However, the most fundamental concern — the mismatch between the recursive-rollout theoretical framework and the single-forward-pass baselines (W1) — was reframed rather than resolved, and this reframing weakens the paper's theoretical contribution from a formal derivation to an analogy. Combined with the still-generic Struct-Aided Predictor motivation (W5) and unaddressed gating claim (W11), the overall assessment does not change sufficiently for an upward revision. I maintain my Overall Recommendation (3) and Confidence (4).

---

> > > ### Author Response · Authors · 2026-04-06
> > >
> > > W1
> > >
> > > Please refer to our round 2 reply to Reviewer 4rgi on Theory.
> > >
> > > W2
> > >
> > > Eq.21 is meaningful under three conditions:
> > >
> > > 1. Small conditional projection residual: Eq.21 is useful when the conditional-expectation projection closely approximates actual rollout behavior. In our above three-forms check, the full achieves consistently higher $R^2$ and lower MAE/RMSE, indicating small projection error where the theory applies.
> > > 2. Bounded local curvature: the Taylor expansion is used in local switching regions with limited state deviation. Under standard smoothness assumptions, second-order terms stay controlled.
> > > 3. Sufficient contraction margin: the estimated coefficients satisfy $\rho \in (0,1)$ across datasets and horizons, implying a non-zero contraction margin $1-\rho > 0$. Small approximation errors act only as perturbations and do not invalidate contraction.
> > >
> > > They indicate Eq.21 is a tight local approximation in practice, supporting the Taylor expansion and contraction-based lag bound.
> > >
> > > W3
> > >
> > > Proof: L-Context ($z_t$) directly estimate the difference ($dx$).
> > >
> > > By construction, the model defines a deterministic mapping $z_t = F(dx_1,\dots,dx_t; x_0),$
> > > where $x_0$ is the initial state of the input. Since $x_t = x_0 + \sum_{i=1}^t dx_i,$
> > > the dependence of $z_t$ on the original inputs can be equivalently expressed through the difference $(dx_1,\dots,dx_t)$ and the initial state $x_0$.
> > >
> > > Since the model consists of smooth neural components, the induced mapping $F$ is locally Lipschitz continuous on compact subsets of the input space. In particular, for any fixed difference sequence $dx$ and any two initial states $x_0, x_0' \in \mathcal{B}$, where $\mathcal{B}$ denotes a bounded set of admissible initial states in the dataset, there exists a constant $L > 0$ such that $\|F(dx; x_0) - F(dx; x_0')\| \le L \|x_0 - x_0'\|.$
> > >
> > > It shows variations in the initial state induce only bounded changes in the $z_t$. Hence, there exists a function $G$ such that $\|z_t - G(dx)\| \le L \cdot \mathrm{diam}(\mathcal{B}),$
> > > which implies the L-Context can be approximated by a function of the difference up to a bounded error. Therefore, L-Context can estimate the difference sufficiently.
> > >
> > > W5
> > >
> > > In our change-aware forecasting tasks, regime switches or pattern changes may occur at arbitrary positions within the input window. If the predictor relies only on absolute positional encoding, it may exploit incidental correlations between change patterns and their locations, learning where changes occur rather than how they evolve. This is problematic since similar patterns may appear at different timestamps at test time, leading to weaker generalization and slower adaptation. This also explains our advantage under standard positional encoding.
> > >
> > > The Struct-Aided Predictor addresses this by introducing a patch-wise relative position basis, providing a location-agnostic structural prior. It encourages the model to focus on relative change patterns rather than absolute time indices. Combined with L-Context, which indicates the current regime, it enables the predictor to align with change dynamics regardless of position. As noted in our first rebuttal, its benefit lies in more stable translation of change information into local dynamic responses, rather than serving as a generic positional refinement.
> > >
> > > W6
> > >
> > > As requested, we added datasets across multiple domains (electricity, weather, and energy) with consistent parameter settings. Results remain consistent: our method adapts faster after changes and reduces error accumulation in switching windows.
> > > https://anonymous.4open.science/r/re2-D/lag.pdf
> > >
> > > W11
> > >
> > > We perform an additional event-conditioned analysis.
> > >
> > > Using externally defined event windows (same as Sec.5.4.3), we analyze all feature channels during test-time forward passes. Since this analysis focuses on the immediate local response of the model, and high-frequency noise introduced by differencing is typically transient and short-lived, we use a short window (size 3) to isolate short-term dynamics.
> > >
> > > Within event and non-event regions, we compare the average gate value r1($E[m \mid event]$-$E[m \mid non\text{-}event]$), and the retained increment ratio r2($\frac{|\Delta x \cdot \text{gate}|}{|\Delta x| + \varepsilon}$).
> > > Across datasets, we observe consistent trends. On ETTh1, in 81.25% of feature-slice pairs (i.e., feature×temporal segment) satisfy r1>0, and 80.36% show higher r2 in event regions. On ETTm2, these proportions are 75.03% and 63.39%, respectively.
> > >
> > > The results show the gate can detect changes and selectively amplifies changes in structured regions, not uniformly reweighting them. It preserves more in event regions and suppresses them in stable, noise-prone areas, aligning with its goal of reducing noise while retaining meaningful changes.
> > >
> > > We thank the reviewer again for the insightful comments and constructive suggestions. We hope that the additional analyses and clarifications help address the main concerns and inform the final evaluation.

---

### Official Review · Reviewer_7QWb · 2026-03-13

**Soundness:** 3
**Presentation:** 3
**Significance:** 3
**Originality:** 3
**Overall Recommendation:** 5
**Confidence:** 4

**Summary:**

The paper proposes a change-aware approach called L-Drive, using L-Context for speeding up the adaptation to changes in the sequence, and also using shared basis functions in the patch for reducing absolute position-dependent perception. The paper also offers a theoretical perspective on the relationship between direct mapping and inertia using EWMA. The paper also conducts experiments using the LTSF and EPF datasets, showing the improvement of prediction using L-Context, as well as the lower computational cost and improved portability of the approach.

**Compliance With Llm Reviewing Policy:**

Affirmed.

**Final Justification:**

The authors have provided detailed responses. They clearly addressed my key concerns (around the specific computational details of L-Context as well as stability), providing well-reasoned explanations and additional analysis. Therefore, I believe the quality of the manuscript has significantly improved, and I support raising the score.

**Key Questions For Authors:**

- The framework describes concatenating the L-Context with the original sequence. Is this a concatenation operation along the time dimension? What is the tensor shape and index in this process?
- Will the differential and gating approaches for processing the L-Context amplify some noise or emphasize some incorrect changes?

**Limitations:**

yes

**Strengths And Weaknesses:**

From my perspective: In soundness, the method is reasonable, and the results are in line with the main conclusions. In theoretical analysis, the method has a certain level of rationality, although it may not have further discussion and related verification in noisy environments. In presentation, the paper is well organized, and the motivation is clear, although the definitions of some symbols and the shape of some tensors are not clear, which has a certain effect on the overall readability. In significance, the paper studies a related problem of prediction lag with the variation of the distribution and proposes a novel experiment on the lag tail metric. In originality, it is based on the perspective of lag reduction and proposes the portability of the baselines, which has a certain level of innovation.

---

> ### Author Rebuttal · Authors · 2026-03-31
>
> We thank the reviewer for the positive feedback and constructive suggestions, and provide our responses below.
>
> Reply to Question1:
>
> We thank the reviewer for this question. The concatenation operation here is not performed along the time dimension, but rather along an additional branch dimension. The specific process is as follows.
>
> Let the original sequence be $X \in {R^{T \times V}}$, and the corresponding L-Context representation be $Z \in {R^{T \times V}}$, where $T$ denotes the sequence length and $V$ denotes the number of variables, which is consistent with the notation in the original manuscript.
>
> First, we apply the same dimension reshaping to both tensors:
>
> $X \to {R^{T \times 1 \times V}},\quad Z \to {R^{T \times 1 \times V}}$
>
> The purpose here is to isolate the time series corresponding to each variable and explicitly introduce an intermediate dimension for subsequent fusion.
>
> Subsequently, we concatenate the two tensors along this newly added dimension, rather than along the time dimension. Thus, we obtain:
>
> $[X;Z] \in {R^{T \times 2 \times V}}$
>
> In other words, for each variable and time step, the concatenated tensor simultaneously retains the original sequence component and the corresponding L-Context component. Therefore, the logical sequence is: the original sequence and the L-Context initially maintain the same shape, are then uniformly reshaped, and finally concatenated along the newly added dimension.
>
> Reply to Question2:
>
> We would like to clarify that the difference and gating mechanism in the L-Context should not be simply interpreted as indiscriminately amplifying noise or erroneously highlighting unrealistic changes. While the difference operation explicitly extracts local change information—which indeed increases the model's sensitivity to short-term fluctuations—this does not imply that the method automatically amplifies noise. The key lies in the fact that these difference signals are not utilized directly; rather, they undergo a gating selection process jointly modeled with the original sequence.
>
> More specifically, the gating weights are determined not solely by the difference signals, but jointly by the original states and the difference information. Therefore, instead of uniformly amplifying all incremental components, this module performs a conditional, selective re-weighting of various change components based on the current observational context. In other words, only those change patterns that exhibit high consistency with the current state are more likely to be retained and passed to subsequent representations. Local perturbations lacking contextual support are not automatically treated as valid changes.
>
> In this sense, our method leverages the gating mechanism to establish a conditional association between the original signals and the change signals, enabling the L-Context to focus on change cues that hold explanatory power for the current dynamics. Consequently, we conceptualize this module as a context-constrained change selection mechanism, rather than a mechanical amplifier of difference results. Compared to directly applying difference signals, our joint gating design minimizes the risk of misidentifying random perturbations as valid changes.
>
> To further demonstrate that the L-Context genuinely captures change information, we have included an additional correlation experiment. We computed the distance correlation (dCor) between the original difference signals and the L-Context representations on held-out data.
>
> This metric evaluates the consistency of their internal change structures along the time dimension. The experimental results reveal a significant structural correlation between the L-Context and the original difference sequence. This confirms that the L-Context successfully retains information related to change patterns; moreover, it indicates that by incorporating the original context, the model selectively extracts and encodes the change structures that best explain the current dynamics.
>
> The full results are as follows:
>
> https://anonymous.4open.science/r/LRe-CE4F/corr.md

---

> > ### Author Rebuttal · Reviewer_7QWb · 2026-04-02
> >
> > The authors have provided detailed responses. They clearly addressed my key concerns (around the specific computational details of L-Context as well as stability), providing well-reasoned explanations and additional analysis. Therefore, I believe the quality of the manuscript has significantly improved, and I support raising the score.

---

### Official Review · Reviewer_4rgi · 2026-03-15

**Soundness:** 2
**Presentation:** 3
**Significance:** 2
**Originality:** 3
**Overall Recommendation:** 3
**Confidence:** 3

**Summary:**

The paper proposes L-Drive, a change-aware forecasting framework that introduces a Latent Context module to address response lag in direct-mapping time series models. The article justifies the proposed method with theoretical findings and empirically with an extensive evaluation. They also performed several analyses of the proposed method, like ablation studies and sensitivity analysis.

**Compliance With Llm Reviewing Policy:**

Affirmed.

**Key Questions For Authors:**

1. Can you demonstrate that L-Context representations correlate with future increments to validate the theoretical connection in Eq. 12?
2. Why is a GRU chosen over a non-recurrent alternative? Have you ablated the sequential integration step specifically? This question can also hold for the other heuristic choices made for building the whole architecture.
3. Have the results been tested for statistical significance? How many reported "best" results survive statistical tests at standard significance levels?

**Limitations:**

Yes

**Strengths And Weaknesses:**

# Strengths
- The paper tackles a genuine and underexplored problem, that is, response lag around regime switches for time series forecasting
- The paper includes theoretical justifications and proofs for their proposed method
- Results are consistent across several experimental settings
- The proposed method is carefully analyzed with ablation and sensitivity studies
# Weaknesses
## Theory and architecture are not tightly coupled
This is the central concern. The theoretical result, which establishes that any auxiliary signal providing a good increment estimate will reduce lag, is a clean and useful result. However, moving from theory to practice introduces a lot of heuristic choices (e.g., standard normalization, 1D convolutions for cross-channel alignment, GRU integration, etc), none of which are derived or justified convincingly, despite being empirically reasonable.
Also, for presentation purposes, the Theoretical Implication section (4.3) appears after the architecture is fully specified, which means it reads as a post-hoc rationalization rather than design guidance. A cleaner structure would be: establish the theoretical desiderata for a good increment estimator, then show that each architectural choice satisfies them.
## Experiments
- The paper reports MSE and MAE exclusively. Both metrics are sensitive to outliers and scale differences across datasets — precisely the conditions that regime switches create. For a paper whose central claim is about behavior at switching points, this is a meaningful concern: a model that is very wrong for a few steps around a switch will accumulate high MSE/MAE disproportionately, and the improvement may reflect better handling of a small number of high-magnitude errors rather than a broadly better forecast. MASE or MAPE would provide scale-normalized alternatives.
- Statistical significance of results is not established. Many reported improvements are in the third decimal place. Results are averaged across forecast horizons without reporting variance, and there is no statistical significance testing. Given that benchmarks are run with fixed seeds and differences are often at the noise level, it is unclear how many of the claimed 12 best-in-class results would survive proper significance testing. This is a significant methodological gap for a paper making broad state-of-the-art claims.
## Presentation issues (minor)
Figure 2 is dense to the point of being uninformative. The diagram attempts to show the full forward pass at once, with overlapping notation and unlabeled intermediate tensors. It does not help the reader understand the data flow; the textual description in Section 4 is clearer. A simplified schematic showing only the high-level information flow would be more useful.
Figure 3 (L-Context visualization) is scientifically weak. The heatmap lacks a colorbar, the quantity being visualized is not precisely defined, and the qualitative alignment between "bright blocks" and "rapid change" is presented as an interpretive observation without any quantitative support. The claim that L-Context "adaptively adjusts context intensity based on stage-wise changes" cannot be verified from this figure alone, and, in general, Section 5.4.2 as a whole does not meet the standard of a scientific analysis.

---

> ### Author Rebuttal · Authors · 2026-03-31
>
> We sincerely thank the reviewer for their careful reading and constructive feedback. We are glad that you appreciated our efforts in addressing the practical problem of regime switches, along with our theoretical proofs, consistent experimental results, and thorough ablation studies. We deeply value your detailed suggestions, which are instrumental in helping us further elevate the quality of this paper.
>
> Below, we response the comments point by point.
>
> Reply on Theory:
>
> We thank the reviewer for this question. We clarify that Section 4.3 is not intended to rigorously derive all architectural details to uniquely determine a specific implementation, but to establish a core theoretical constraint: if the auxiliary representation provides more stable and lower-error increment-related information, both tracking error and residual lag will be reduced. From this perspective, the modules in our method are not arbitrary, but implementation choices guided by this constraint. For example, normalization improves the numerical stability of incremental signals; 1D convolutions help align cross-channel variation patterns within local temporal windows; and GRU integrates temporal dynamics. All components serve the same goal: effective extraction and integration of increment-related information.
>
> Under this principle, a key empirical question is whether L-Context indeed captures increment-related information. To verify this, we conduct an additional experiment by computing the distance correlation(dCor) between the raw difference signal and the L-Context representation on held-out data. Specifically, we compute pairwise distances across time steps and measure the correlation between the resulting distance structures; values closer to 1 indicate stronger correlation. This evaluates whether the intrinsic variation structure of the difference sequence is preserved in the L-Context space.
>
> The averaged results over all prediction lengths are as follows:
> | Datasets | dCor_Mean | dCor_Std |
> |----------|----------:|---------:|
> | ETTh1    | 0.838     | 0.003    |
> | ETTh2    | 0.806     | 0.003    |
> | ETTm1    | 0.800     | 0.006    |
> | ETTm2    | 0.794     | 0.005    |
> | Weather  | 0.782     | 0.006    |
> | Solar    | 0.880     | 0.004    |
> | EPF      | 0.892     | 0.002    |
>
> The full results are as follows:
>
> https://anonymous.4open.science/r/LRe-CE4F/corr.md
>
> Across repeated experiments, the correlation remains high across all datasets (0.782–0.892), with consistently small standard deviations, indicating strong and stable structural correlation between the raw difference signal and L-Context. This suggests that L-Context indeed encodes variation-related information. We believe this provides direct empirical support for interpreting L-Context as an increment-related signal in Eq.(12).
>
> Reply on Experiment:
>
> 1. In response to the reviewer, we added the MAPE metric and compared against the two latest baselines in the main text, while keeping their original parameter settings unchanged. Our method still achieving superior performance on most prediction lengths. Notably, on EPF, the maximum average improvement over all baselines reaches 25.64%. The full results are as follows.
>
> https://anonymous.4open.science/r/LRe-CE4F/MAPE.md
>
> 2. For the main experiments and the two key ablations, we report mean and standard deviation over multiple runs. The full results are as follows.
>
> https://anonymous.4open.science/r/LRe-CE4F/std.md
>
> Overall, the standard deviations remain low across all datasets and prediction horizons, demonstrating the stability and consistency of the proposed model over repeated runs. In comparison, the ablated variants show slightly increased fluctuations in standard deviation in certain cases.
>
> Reply to Minor Issues:
>
> We clarify that Figure 3 is not intended as a standalone mechanism validation, but as an intuitive supplement to the L-Context representation pattern, illustrating its non-uniform responses across temporal stages. We will revise the description and improve the figure for clarity in the revision.
>
> Once again, we sincerely thank you for your time and all your valuable comments. We hope these additional experiments and clarifications address the main concerns and help reassess the contribution of the paper.

---

> > ### Author Rebuttal · Reviewer_4rgi · 2026-04-03
> >
> > I thank the authors for their thorough rebuttal and the additional experiments conducted. The inclusion of the dCor computation, along with the reported means and standard deviations, represents a meaningful improvement to the paper.
> >
> > That said, my primary concerns persist and center on two interrelated issues: the accumulation of design choices in the architecture and the rigor of the experimental evaluation. As Reviewer nF9B has also noted, the theoretical grounding relies on several approximations that have not been examined in sufficient depth, making the contribution closer to an analogy than a formal theoretical result. I understand that Section 4.3 is not intended as a rigorous derivation of every architectural detail. However, the variety of design decisions still weakens the claimed link between theory and practice. Moreover, the architectural complexity and the volume of added components widen this gap further and make it particularly difficult to conduct scientifically sound experiments — especially when the observed improvements are marginal.
> >
> > In particular, my most pressing concern lies with the experimental protocol, specifically the handling of hyperparameter tuning. Time series forecasting is notoriously sensitive to dataset characteristics, prediction horizons, and history lengths. Simply increasing the number of architectural parameters can, on its own, shift results. I would welcome further details on how hyperparameters were selected. Adopting hyperparameter configurations directly from baseline methods, for example, is insufficiently rigorous, particularly given the narrow performance margins reported. If the authors address this point convincingly, I would be glad to reconsider my score.

---

> > > ### Author Response · Authors · 2026-04-06
> > >
> > > # Theory
> > >
> > > We provide a consolidated clarification of the theoretical logic across both the main paper and appendix, to better position the theory and address concerns about the derivation, mechanism, and practical relevance.
> > >
> > > Eq.2 is not the literal inference graph. It is a local reduced-form description of the prediction inertia.(addressed in round1)
> > > Eq.18–22 derive the reduced-form operator from a general predictor (with Eq.21 defining the conditional expectation due to state complexity; addressed in round1). Eq.23–30 then obtain Eq. 2 via exact decomposition and first-order Taylor expansion.
> > >
> > > Together with the local contraction conditions in Eq.27–30, this gives $\left| {{\rho _t}} \right| \leqslant L < 1$. This ensures contractive, stable (non-divergent) inertial dependence in Eq.2, validating the reduced form and establishing the theoretical link from a general predictor to a local reduced form.
> > >
> > > We further evaluate how well the baselines satisfy the reduced form, addressing Reviewers’ concern:
> > > We collect sliding-window rollout predictions, detect switching events from ground truth, and construct local neighborhoods (same as Sec.5.4.3). To avoid leakage, we use event-level train/test splits.
> > >
> > > We fit three forms:
> > >
> > > 1.Inertia-only: prediction depends on its previous output, capturing prediction inertia.
> > >
> > > 2.Update-only: prediction depends only on window updates (newly entered and removed elements), both of which are necessary due to the sliding-window mechanism.
> > >
> > > 3.Full (Eq.(2) proxy): combines both, corresponding to Eq. (2) and approximating rollout behavior.
> > >
> > > Metrics: $R^2$: variance explained, measuring approximation quality, higher is better; MSE / MAE: fitting error, lower is better; $\rho$: inertia strength.
> > >
> > > https://anonymous.4open.science/r/re2-D/form.pdf
> > >
> > > Results:
> > >
> > > 1. Inertia is strong but insufficient: the inertia-only model achieves high $R^2$, confirming strong dependence on past predictions, but is consistently outperformed by the full model.
> > >
> > > 2. Update alone is unstable: the update-only shows lower $R^2$ and increased MAE/RMSE.
> > >
> > > 3. The full consistently achieves the best performance: it improves $R^2$ while reducing MAE/RMSE, indicating that rollout behavior is best captured by combining inertia and updates.
> > >
> > > In summary, both theory and quantitative results support Eq.2 is a faithful local approximation of baseline rollout behavior, rather than a conceptual analogy.
> > >
> > > We further clarify why increment is key and why L-Context captures it. As noted in the paper, before a switch, a baseline cannot distinguish whether the current regime persists or a new one begins. Under fixed lookback, new-regime information enters only gradually. Thus, modeling the increment is key to faster incorporation of new-regime information. Moreover, the correlation proof in our round2 response to Reviewer nF9B(W3) and the round1 dCor verify the link between L-Context and the increment, consistent with both theory and observation.
> > >
> > > Under this view, Section 4.3 should be interpreted as a conditional lag bound: given that the baseline near switching points is well approximated by the above local reduced form, the role of L-Context and its error bound follow rigorously.
> > > Thus, the theory is self-consistent.
> > >
> > > # Protocol
> > >
> > > We clarify the fairness of hyperparameter tuning as follows. For all baselines, we used the recommended settings from their original papers or official codebases, following a commonly adopted practice in prior work such as iTransformer (ICLR 2024 Spotlight). Our method also uses the same data split, training/evaluation protocol and input length(96) as the baselines, and tuning is restricted to standard hyperparameters (e.g., batch size, learning rate, hidden dimension). So all methods are evaluated under similar tuning budgets rather than with extra tuning effort for ours.
> > >
> > > To further address the reviewer’s concern, we additionally adjusted the baselines so that their parameter-count gap remain within 10% of ours, and report mean/std over multiple runs. Consistent with ours, the adjustment was restricted to standard hyperparameters only. Results (URL below) show our method still performs better, suggesting the gains cannot be explained by parameter count alone.
> > >
> > > https://anonymous.4open.science/r/re2-D/para.pdf
> > >
> > > Moreover, the added modules are lightweight: in L-Context, the gating and GRU dimensions are matched to the input, and in the Struct-Aided Predictor, the relative positional bases dimension is fixed to 2 across datasets. We also provide sensitivity analyses(see URL below) showing stable performance across reasonable hyperparameter ranges, which further support improvements come from the proposed design rather than tuning advantage or brute-force complexity.
> > >
> > > https://anonymous.4open.science/r/re2-D/result.pdf
> > >
> > > Once again, we sincerely thank you for your time and all your valuable comments. We respectfully hope that these additional analyses and clarifications will be considered in the final assessment.

---

### Official Review · Reviewer_SEF1 · 2026-03-24

**Soundness:** 4
**Presentation:** 4
**Significance:** 3
**Originality:** 3
**Overall Recommendation:** 5
**Confidence:** 4

**Summary:**

This research addresses the response lag problem in time series forecasting during periods of change by proposing the L-Drive based on L-Context. Furthermore, this model employs a Struct-Aided Predictor across multiple patches to better capture local structure in response to changes.

**Compliance With Llm Reviewing Policy:**

Affirmed.

**Final Justification:**

After reading the rebuttal, I find that the authors have fully addressed my main concerns, and the clarifications help improve my understanding of the work. Overall, the rebuttal largely alleviate my concerns, and I therefore increase my score.

**Key Questions For Authors:**

1. What considerations underlie the choice of the difference order? Do different scales of difference exhibit different performance characteristics?
2. Are the experimental metrics TailAUC and ExcessAUC proposed for addressing lag robust?
3. What is the difference between this article and related work on potential states?

**Limitations:**

Although the article mentions some research directions for future work, such as conducting more fine-grained change modeling and robustness research in the broader real world, the discussion of limitations in the article is still somewhat insufficient.

**Strengths And Weaknesses:**

1. This research argues that Direct-Mapping methods may exhibit response lag during periods of fluctuation, and uses this as a key point for improvement. It first provides a mechanism explanation based on inertial recursion, and then proposes an L-Context centered on the differential signal, supported by various experiments. The overall logic is relatively coherent, but it lacks analysis on the choice and order of the differential signal, and the lag error indices TailAUC and ExcessAUC lack robustness evidence.
2. It is clearly structured and includes code, which allows readers to understand the method. However, some details need to be supplemented by the author. For example, the process of combining L-Context and the original sequence in Section 4.2 lacks a more detailed description.
3. Using the prediction error within the variation range as the diagnostic target and proposing structural measures, the article includes relevant experiments comparing efficiency and performance, which has some practical significance. However, the evidence related to lag lacks sensitivity experimental analysis, such as thresholds, and it is recommended that the authors add further analysis to make the experimental evidence more reliable.

---

> ### Author Rebuttal · Authors · 2026-03-31
>
> We thank the reviewer for the positive feedback and valuable suggestions, and provide our responses below.
>
> Q1:
>
> We adopted the first-order difference primarily based on the following considerations:
>
> 1. Methodological Objective: Our focus is on the local change information around regime switches. The first-order difference most directly characterizes the incremental changes between adjacent time steps, making it highly consistent with our theoretical analysis of increment-related information.
>
> 2. Signal Properties: While enhancing sensitivity to changes, the first-order difference preserves good local interpretability. Compared to higher-order or larger-interval differences, it generally avoids prematurely introducing excessive oscillation amplification and numerical instability.
>
> In other words, we selected the first-order difference to achieve a robust balance among change sensitivity, stability, and interpretability.
>
> Regarding the performance of differences at various scales, we agree that they would indeed exhibit distinct characteristics. Generally, smaller-scale differences are more sensitive to short-term local changes, making them ideal for capturing sudden switches or rapid slope variations. Conversely, larger-scale or higher-order differences might be more sensitive to slower structural changes, but they are also more prone to amplifying noise, weakening local correspondences, and complicating the subsequent gating and context modeling.
>
> Q2:
>
> 1. On Metric Design:
>
> The event detection here is not intended to replicate the internal mechanism of L-Context, but rather to provide a model-agnostic external definition for the "change segment." By aggregating first-order differences on the ground-truth series, we merely flag moments with high local change intensity as evaluation events. This allows us to measure whether the error tails of different methods become more severe within the change neighborhood. In other words, this analysis focuses on the error concentration around externally detected change regions, rather than testing whether a specific model is better at utilizing first-order difference features. Therefore, this evaluation is not a self-serving metric constructed based on our model's internal outputs.
>
> 2. On the Selection of Baselines:
>
> Regarding the selection of baseline models in Table 4, we would like to clarify that the purpose of this analysis is not to provide an exhaustive display of all baselines. Instead, we selected a diverse subset of models to compare their lag-related error behavior. As observed in the Figure 1 example of our manuscript, under scenarios with significant abrupt changes, the error trailing patterns exhibited by CrossLinear and iTransformer within the switching neighborhood are broadly similar and show some overlap. Consequently, we retain only one of them for reporting in the main text. We will also include the corresponding results in the revised version.
>
> 3. On the Details of Lag Error Parameters:
>
> The Appendix G includes the detailed procedures for the Lag Error Analysis, covering event detection, as well as the definitions and calculation methods for TailAUC and ExcessAUC. Furthermore, we would like to add the following details here: We use 25 as the neighborhood window size and set the minimum gap to 24. This configuration is designed to align the granularity of event detection with the time scope of the lag-tail analysis. The former calculates the error tail within the local window immediately following a change, while the latter prevents double-counting the same change segment when a high change intensity persists for several steps. Because the minimum gap is essentially aligned with the neighborhood window, the evaluation windows of adjacent events do not significantly overlap. Thus, this setting stably captures the extra error accumulation around change. It is important to note that the aforementioned parameter settings are kept strictly consistent across all baseline models and were not individually tuned or biased in favor of our proposed method.
>
> Importantly, TailAUC and ExcessAUC are not defined by a single instantaneous error point, but by the cumulative error area within the change neighborhood; ExcessAUC further deducts the steady-state baseline of the non-event regions. Structurally, these two metrics emphasize continuous lag-tail behavior over occasional local spikes, thereby enhancing robustness against single-point fluctuations and transient anomalies. In short, the goal of this evaluation protocol is not to amplify error differences at isolated moments, but to robustly characterize the overall intensity of the error tail during a localized period following a change.
>
> Q3:
>
> As discussed in the sixth paragraph of the Introduction, L-Context is not a classical generative latent state and does not rely on explicit state-space equations. Instead, it acts as a prediction-oriented conditional variable for faster adaptation under regime switching.

---

> > ### Author Rebuttal · Reviewer_SEF1 · 2026-04-02
> >
> > Thank you to the authors for their careful rebuttal. The rebuttal has addressed my concerns and helped improve the clarity of the paper. I decide to increase my socres. Some of these points could be incorporated into the revision to further improve the clarity of the paper.

---

### Decision · Program_Chairs · 2026-04-30

**Decision:**

Accept (regular)

**Comment:**

This paper identifies response lag in direct-mapping forecasting models during distribution fluctuations, provides a mechanistic explanation based on inertial recursion, and proposes L-Context centered on differential signals. The method is portable across architectures. The problem formulation is a strong contribution: explicitly separating lag from general distribution shift. Theoretical derivations are internally consistent, and experiments include ablation, sensitivity, and tail metrics.

All four reviewers now recommend acceptance. In the discussion, the initially negative reviewer confirmed that their concerns have been adequately addressed through the rebuttal and supplementary analyses. Remaining issues are minor wording/framing improvements. The authors are recommended to carefully revise the final version.